# Numerical coupling of aerosol emissions, dry removal, and turbulent mixing in the E3SM Atmosphere Model version 1 (EAMv1), part I: dust budget analyses and the impacts of a revised coupling scheme

**Hui Wan**[1], **Kai Zhang**[1], **Christopher J. Vogl**[2], **Carol S. Woodward**[2], **Richard C. Easter**[1], **Philip J. Rasch**[3], **Yan Feng**[4], **and Hailong Wang**[1]

[1]Atmospheric, Climate, and Earth Sciences Division, Pacific Northwest National Laboratory, Richland, Washington, USA
[2]Center for Applied Scientific Computing, Lawrence Livermore National Laboratory, Livermore, California, USA
[3]Department of Atmospheric Sciences, University of Washington, Seattle, Washington, USA
[4]Environmental Science Division, Argonne National Laboratory, Lemont, Illinois, USA

**Correspondence:** Hui Wan (Hui.Wan@pnnl.gov)

**Abstract.** An earlier study evaluating dust life cycle in the Energy Exascale Earth System Model (E3SM) Atmosphere Model version 1 (EAMv1) has revealed that the simulated global mean dust lifetime is substantially shorter when higher vertical resolution is used, primarily due to significant strengthening of dust dry removal in source regions. This paper demonstrates that the sequential splitting of aerosol emissions, dry removal, and turbulent mixing in the model's time integration loop, especially the calculation of dry removal after surface emissions and before turbulent mixing, is the primary reason for the vertical resolution sensitivity reported in that earlier study. Based on this reasoning, we propose a revised numerical process coupling scheme that requires the least amount of code changes, in which the surface emissions are applied before turbulent mixing instead of before dry removal. The revised scheme allows newly emitted particles to be transported aloft by turbulence before being removed from the atmosphere, and hence better resembles the dust life cycle in the real world.

Sensitivity experiments show that the revised process coupling substantially weakens dry removal and strengthens vertical mixing in dust source regions. It also strengthens the large-scale transport from source to non-source regions, strengthens dry removal outside the source regions, and strengthens wet removal and activation globally. In transient simulations of the years 2000 to 2009 conducted using 1° horizontal grid spacing, 72 vertical layers, and unchanged tuning parameters of emission strength, the revised process coupling leads to a 40% increase in the global total dust burden and an increase of dust lifetime from 1.8 days to 2.5 days

in terms of 10-year averages. Weakened dry removal and increased mixing ratios are also seen for other aerosol species that have substantial surface emissions, although the changes in mixing ratio are considerably smaller for the submicron species than for dust and sea salt.

Numerical experiments confirm the revised coupling scheme significantly reduces the strong and nonphysical sensitivities of model results to vertical resolution in the original EAMv1. This provides a motivation for adopting the revised scheme in EAM as well as for further improvements on the simple revision presented in this paper.

## 1 Introduction

The numerical modeling of aerosol-climate interactions is a research topic with high levels of uncertainty (Seinfeld et al., 2016; Bellouin et al., 2020; Smith et al., 2020). State-of-the-art global aerosol-climate models (see lists in, e.g., Gliß et al., 2021 and Myhre et al., 2013) often consider a fairly large number of physical and chemical processes that affect aerosol life cycles. Such processes include emissions, new particle formation, particle growth and aging, transport by winds at local to global scales, as well as removal from the atmosphere due to gravitational settling, boundary layer processes, formation of clouds and precipitation, and wet removal by precipitation. Typically, the numerical representations of different processes are separately developed by subject matter experts and subsequently assembled to form

global models. Different aerosol-climate models are known to show substantial discrepancies in the simulated magnitudes and spatiotemporal variations of the source and sink processes (Textor et al., 2006; Gliß et al., 2021). Continuous efforts are being made to attribute such discrepancies to the physical assumptions and computational methods used for individual processes, while investigations on the impact of numerical process coupling (e.g., Wan et al., 2013) are relatively rare.

Process splitting is a ubiquitous method used in the numerical simulation of multiphysics problems. In the pollution modeling community and in the development of chemical transport models (CTMs), abundant literature exists on the numerical techniques for solving equations using splitting methods. Here we do not attempt to provide a comprehensive or representative list of references but instead simply mention a couple of examples of mathematical analysis (Lanser and Verwer, 1999; Dimov et al., 2008) and provide pointers to the concrete atmospheric chemistry and physics examples discussed in Chapters 16 to 19 in Jacobson (2005) and Chapter 25 in Seinfeld and Pandis (2016). In the literature on climate models used for understanding aerosol-climate interactions and predicting aerosol-induced climate changes, however, detailed documentation and discussions on numerical coupling are much harder to find. For example, a list of primary references for the models participating in the AeroCom (Aerosol Comparisons between Observations and Models) Phase III intercomparison can be found in Table 2 and Sect. S4 of Gliß et al. (2021). These references include Matsui (2017), Matsui and Mahowald (2017), van Noije et al. (2014), van Noije et al. (2021), Tegen et al. (2019), Bergman et al. (2012), Kokkola et al. (2018), Rémy et al. (2019), Simpson et al. (2012), Colarco et al. (2010), Zhao et al. (2018), Bauer et al. (2020), Tsigaridis et al. (2013), Koch et al. (2007), Koch et al. (2006), Balkanski et al. (2004), Schulz et al. (2009), Seland et al. (2020), Kirkevåg et al. (2018), Lund et al. (2018), Søvde et al. (2012), Takemura et al. (2009), and Takemura et al. (2005). This collection of models, as an example subset of more models of similar kind, include not only climate models but also weather forecast models and CTMs. The above-listed references contain comprehensive descriptions of the computational representation of aerosol particle size distributions as well as the modeling assumptions used in the parameterization of various physical and chemical processes. In some of these papers, descriptions or summaries are provided for the numerical treatments of some individual processes, see, e.g., Sects. 2.1.1 and 2.1.2 in Matsui (2017), Sect. 2.4 in Bergman et al. (2012), Table 1 in Kokkola et al. (2018), Sect. S2 in Simpson et al. (2012). In some papers, numerical coupling is documented or commented on for some subsets of processes, see, e.g., Sect. 2.4 in Bergman et al. (2012), Table 1 in Kokkola et al. (2018), Sects. 2.1 and 5.1 in Rémy et al. (2019), Sect. 2 in Koch et al. (2006), Sect. 2.2 in Kirkevåg et al. (2018), and Sect. 2 in Søvde et al. (2012). However, more complete overviews

of the sequence of calculations of the aerosol-affecting processes, similar to the schematic in Fig. 1 of Morcrette et al. (2009) and the compact description in Sect. 2.3 of Huijnen et al. (2010), are generally not included but would be informative.

The contrast between the abundant literature on splitting algorithms in the CTM and pollution modeling communities and the much rarer discussions on this topic in global aerosol models seems consistent with a statement in the book by Glowinski et al. (2016) that "practitioners of the above [splitting] methods have become quite specialized, forming subcommunities with very few interactions between them." For the development of weather, climate, and Earth system models, the review paper by Gross et al. (2018) has pointed out that numerical process coupling is a largely overlooked topic deserving more attention. Recent studies by, e.g., Donahue and Caldwell (2018), Barrett et al. (2019), Donahue and Caldwell (2020), Wan et al. (2021), Santos et al. (2021), Ubbiali et al. (2021), and Zhou and Harris (2022) have presented various efforts on identifying and addressing some of the numerical coupling issues related to the simulation of clouds and general circulation. Here, we present an example of a numerical issue related to aerosols.

The Energy Exascale Earth System Model (E3SM) is an Earth system model developed by the U.S. Department of Energy for addressing science questions related to the prediction of Earth system dynamics and climate change (Leung et al., 2020; E3SM Project, 2018). The E3SM Atmosphere Model version 1 (EAMv1, Rasch et al., 2019) is an atmospheric general circulation model that includes a comprehensive representation of the life cycles of various natural and anthropogenic aerosol species. A recent study by Feng et al. (2022) evaluated the dust life cycle in EAMv1 simulated with different horizontal and vertical resolutions. It was shown that an EAMv1 simulation using $1°$ horizontal grid spacing and 72 vertical layers produced a global mean dust lifetime of 1.85 days, which was substantially shorter than the lifetime of 2.6 days reported by Liu et al. (2012) and Scanza et al. (2015) who used the predecessor model CAM5, the Community Atmosphere Model version 5 (Neale et al., 2012), with $2°$ horizontal grid spacing and 30 layers. Feng et al. (2022) also showed that reverting EAMv1's vertical resolution from 72 layers to 30 layers would lengthen the global mean dust lifetime from 1.85 days to 2.4 days (see Table 4 therein), primarily through the weakening of dry removal in the dust source regions.

This paper investigates the vertical resolution sensitivities of dust lifetime and dry removal reported in Feng et al. (2022). Sections 2.1 and 2.2 provide a brief overview of EAMv1 and its parameterizations of aerosol emissions, dry removal, and turbulent mixing. Section 2.3 describes EAMv1's numerical scheme used for coupling the aerosol-related processes and compares it with the treatments used in some other models. Section 3 analyzes the simulated dust mass budget and compares simulations conducted with 72

or 30 layers to reveal weaknesses of the numerical process coupling in the publicly released EAMv1 (which we refer to as the original or default EAMv1 in this paper). A simple revision to the numerical process coupling is proposed in Sect. 3.3 and its impacts on the simulated aerosol climatology are evaluated in Sect. 4. The conclusions are drawn in Sect. 5.

As is shown in Sect. 4, the revised process coupling significantly weakens the dust dry removal in the source regions and substantially reduces sensitivities of the simulated dust dry removal rate and lifetime to vertical resolution. These changes appear to be desirable given the deficiencies of the EAMv1 results pointed out in Feng et al. (2022). On the other hand, the revision also results in large, global and systematical increases in the mass burden of dust and sea salt, meaning the top-of-the-atmosphere energy fluxes become out of balance. While such aerosol burden and energy flux changes can be offset by retuning uncertain parameters used in the dust and sea salt emission parameterizations, one might ask whether the revised coupling is more accurate in a numerical sense and whether the retuning is worthwhile. Other than presenting in Sect. 3 some process-level analyses of EAMv1 results to motivate the revision and confirming in Sect. 4 that the effects of the revised coupling on EAM's aerosol climatology meet our expectation, it is not straightforward to obtain additional evidence from EAM simulations to show that the revised coupling is an improvement in the numerical sense. This challenge has to do with the fact that the current EAM code does not allow for convergence testing of numerical process coupling between aerosol emissions, dry removal, and turbulent mixing without changing the timestep sizes and coupling frequencies of other physical processes such as the resolved fluid dynamics and the parameterized cloud processes. To address this challenge, the companion paper by Vogl et al. (2023) shows from an applied mathematics perspective that the local truncation error in dust dry removal caused by process splitting is smaller when the revised scheme is used, hence further justifying the adoption of the revised scheme in EAMv1.

## 2   The EAMv1 model and simulations

The EAMv1 configuration described in Rasch et al. (2019) and used in this study is a global hydrostatic atmospheric model that simulates the spatial distribution and time evolution of air temperature, pressure, winds, humidity, clouds, and precipitation. In addition, the model has 50 prognostic variables corresponding to the mixing ratios of aerosol particles of different sizes (diameters), chemical compositions, and attachment states (interstitial or cloud-borne). The mixing ratios of a few chemical gas species that are precursors of aerosols are also simulated using prognostic equations.

### 2.1   EAMv1 overview

EAMv1's dynamical core solves the primitive equations of the global atmospheric flow using a continuous Galerkin spectral-element method on a cubed-sphere horizontal mesh (Dennis et al., 2012; Taylor et al., 2009). The vertical discretization uses a semi-Lagrangian scheme and a pressure-based terrain-following coordinate (Lin, 2004). The resolved-scale tracer transport uses the discretization method of Lin (2004) but has been adapted to the cubed sphere. The transport algorithm ensures local conservation of tracer and air masses as well as moist total energy (Taylor, 2011).

For the parameterization of unresolved processes, the transfer of solar and terrestrial radiation is calculated with the Rapid Radiative Transfer Model for General Circulation Models (RRTMG, Iacono et al., 2008; Mlawer et al., 1997). Deep convection is parameterized with the scheme of Zhang and McFarlane (1995) with modifications by Neale et al. (2008) and Richter and Rasch (2008). Shallow convection, turbulence, and stratiform cloud macrophysics are represented by the higher-order closure parameterization named Cloud Layers Unified By Binormals (Larson, 2017; Larson and Golaz, 2005; Golaz et al., 2002; Larson et al., 2002). Stratiform cloud microphysics is represented with a two-moment parameterization with prognostic equations for the mass and number concentrations of cloud droplets, ice crystals, rain, and snow (Gettelman and Morrison, 2015; Gettelman et al., 2015; Morrison and Gettelman, 2008). More detailed descriptions of the parameterization suite can be found in Section 2 of Rasch et al. (2019) and the references therein.

### 2.2   Aerosol processes in EAMv1

The Modal Aerosol Module (MAM, Wang et al., 2020; Liu et al., 2016, 2012; Ghan and Easter, 2006; Easter et al., 2004) is a suite of parameterizations developed for global climate modeling, aiming at providing a simplified yet sophisticated representation of aerosol life cycles as well as their interactions with clouds and precipitation. Seven aerosol species are considered in EAMv1: sulfate, black carbon (BC), primary organic aerosols (POA), secondary organic aerosols (SOA), marine organic aerosols (MOA), dust, and sea salt. Mass and number concentration changes are predicted for aerosol particles of two attachment states, interstitial and cloud-borne, which refer to the particles found outside and within cloud droplets, respectively. The two attachment states correspond to two populations of aerosol particles. In the 4-mode configuration of MAM used in EAMv1 (i.e., MAM4), the particle size distribution in each population is represented by one coarse mode and three fine-particle modes, see Fig. 2 in Wang et al. (2020). Within a mode, the particle size distribution is represented by a lognormal function of particle diameter, assuming the particles are spherical and the geometric standard deviation of the lognormal function is fixed. Under these assumptions, the mass mixing ratios of differ-

ent species in different modes, as well as the particle number mixing ratios of the different modes, are predicted.

Aerosol processes currently considered in MAM4 include emissions as well as particle mass and number changes resulting from new particle formation (aerosol nucleation), condensation and evaporation of chemical species, water uptake, coagulation, aqueous chemistry in cloud droplets, aerosol activation (cloud nucleation), resuspension from evaporating cloud droplets and precipitation, in-cloud and below-cloud wet removal, sub-grid vertical transport by deep convection and turbulence, gravitational settling, and turbulent dry deposition. Descriptions of MAM4 and its predecessors can be found in Rasch et al. (2019), Wang et al. (2020), Liu et al. (2016), Liu et al. (2012), Ghan and Easter (2006), and Easter et al. (2004). Here we only briefly summarize the processes that are the foci of this study.

### 2.2.1 Emissions

Since MAM considers a variety of sizes, species, and origins of aerosol particles, a comprehensive set of assumptions and treatments are needed to specify aerosol emissions, see Section S1.1.1 in Liu et al. (2012). In MAM4, the prescription of anthropogenic aerosol mass emissions follows protocols of model intercomparision projects and published studies. The partitioning of the mass emissions into MAM's lognormal modes and the calculation of aerosol number emissions in those modes are based on assumed emission size distributions. Natural aerosol emissions are calculated online (i.e., during a simulation) using emission parameterizations. The scheme from Zender et al. (2003) is used for dust. The calculation of sea salt emissions follows Mårtensson et al. (2003) for particle diameters from 20 nm to 2.5 $\mu$m and Monahan et al. (1986) for particle diameters from 2.5 $\mu$m to 10 nm. For MOA, the parameterization of Burrows et al. (2014) is used to calculate the mass portion of MOA in the emitted sea spray aerosols; this information is used in combination with the predicted sea salt emissions to determine MOA emissions, assuming that MOA emissions add to the sea spray aerosol emissions, and that the emitted MOA is internally mixed with other aerosol species (Burrows et al., 2022).

Dust aerosols are assumed to be emitted only at the Earth's surface. The emission fluxes are parameterized as a function of various properties of the Earth's surface (e.g., soil moisture and erodibility) and atmospheric conditions (e.g., friction velocity and 10 m wind speed). Further details can be found in Section 2.4 of Zhang et al. (2016) and the references therein. In mathematical terms, the emission of dust aerosols results in source terms in the evolution equations of dust mixing ratios; these source terms are non-zero only at the bottom boundary of an atmosphere column. Further comments on this assumption can be found in Sect. 2.4.

### 2.2.2 Turbulent dry deposition and gravitational settling

In this paper, we use the term "dry removal" to refer to both the gravitational settling and the turbulent dry deposition of aerosol particles, as the two processes are handled together in the MAM4 parameterization suite embedded in EAMv1. Gravitational settling is the downward movement of particles under the action of gravity, whereas turbulent dry deposition refers to the loss of particles to the Earth's surface through Brownian diffusion, impaction, interception, etc. For both processes, the downward aerosol mass or number fluxes per unit time across a unit area at the Earth's surface are calculated using the formula

$$M_{i,\mathrm{sfc}} = \rho_b \, q_{i,b} \, v_{i,b} \,. \tag{1}$$

Here, $M_{i,\mathrm{sfc}}$ is the downward flux of the $i$-th aerosol tracer, $\rho_b$ is air density in the bottom layer of the atmosphere (i.e., the lowest model layer above the Earth's surface), and $q_{i,b}$ is the mixing ratio of aerosol tracer $i$ in the bottom layer. $v_{i,b}$ is the downward deposition velocity of aerosol tracer $i$ calculated using the aerosol properties and ambient conditions in the bottom layer.

Like in many other models (see, e.g., Mann et al., 2010; Zhang et al., 2012), MAM4 in EAMv1 assumes gravitational settling of aerosols can occur throughout the atmosphere column. The resulting fluxes at altitudes above the Earth's surface are parameterized as

$$M_i^{\mathrm{grav}}(z) = \rho(z) \, q_i(z) \, v_i^{\mathrm{grav}}(z), \tag{2}$$

where $z$ is geopotential height; $\rho(z)$, $q_i(z)$, $v_i^{\mathrm{grav}}(z)$, and $M_i^{\mathrm{grav}}(z)$ are air density, aerosol mixing ratio, gravitational settling velocity, and gravitational settling flux, respectively, at altitude $z$. The calculation of the settling velocity is based on the Stokes' law, assuming particles reach their terminal velocities instantly. The settling velocity of a single particle is calculated with Eqs. (2) and (3) in Zhang et al. (2001). Correction factors are included to account for the impact of the lognormal size distribution.

The turbulent dry deposition velocity is parameterized using Eq. (21) in Zender et al. (2003), for which the calculation of the quasi-laminar layer resistance follows Sect. 2 in Zhang et al. (2001).

### 2.2.3 Turbulent mixing and activation–resuspension

Turbulent mixing of aerosols in MAM4 is parameterized using the eddy diffusivity approach (see, e.g., Garratt, 1994), which gives aerosol concentration tendencies in the form of

$$\left(\frac{\partial \rho q_i}{\partial t}\right)^{\mathrm{turb\text{-}mix}} = \frac{\partial}{\partial z}\left(\rho K_h \frac{\partial q_i}{\partial z}\right), \tag{3}$$

where $\rho$ is air density, $q_i$ is mixing ratio of aerosol tracer $i$, $z$ is geopotential height, and $K_h$ is eddy diffusion coefficient. In EAMv1, $K_h$ is calculated by the turbulence and

cloud parameterization CLUBB, while the turbulent mixing of aerosol mass, aerosol number, and cloud droplet number is treated separately (outside CLUBB) in conjunction with aerosol activation and resuspension from evaporating cloud droplets. In other words, MAM4's parameterization of turbulent mixing and aerosol activation-resuspension solves differential equations in the form of

$$\left(\frac{\partial \rho q_i}{\partial t}\right)^{\text{turb-mix+act/res}} = \frac{\partial}{\partial z}\left(\rho K_h \frac{\partial q_i}{\partial z}\right) + \left(\frac{\partial \rho q_i}{\partial t}\right)^{\text{act/res}},$$
(4)

where the last term in Eq. (4) is the rate of change caused by aerosol activation and resuspension from cloud droplets. Additional information about the parameterization can be found in Section S1.1.8 in the supplementary materials of Liu et al. (2012) and in Ghan and Easter (2006).

For clarification, we note that MAM also accounts for the resuspension of aerosols from evaporating precipitation as a part of aerosol wet removal. In this paper, when resuspension is mentioned in conjunction with activation, we are referring to the resuspension from cloud droplets.

## 2.3 Numerical process coupling in EAMv1

The schematic in Fig. 1 depicts the sequence in which the various atmospheric processes are calculated within a time window of 30 minutes in the 1° EAMv1 simulations. The schematic also shows where the coupling between EAM and the other components of E3SM (e.g., land and ocean) occurs during the 30-minute time window. EAMv1 uses primarily the sequential splitting method for the numerical coupling of aerosol-related processes and most other parameterizations. With this method, the rates of change (i.e., tendencies) of aerosol mixing ratios caused by a process (or a process group) are calculated and then applied immediately to obtain new (updated) values of the mixing ratios. The updated mixing ratios are then passed on to the next process or group to calculate the next set of tendencies and further update the mixing ratios. Within a 30-minute time window in EAMv1, the aerosol-related processes are calculated as follows. (The numbering below matches the labels in Fig. 1.)

1. Emission fluxes of aerosols are calculated based on atmospheric and Earth surface conditions or derived from emission datasets. Dust emissions are calculated in E3SM's land model and passed to EAM by the coupler. Emissions of the other species are calculated (or read in from data files, partitioned to MAM's lognormal modes, and mapped to EAM's vertical grid) within the atmosphere model.

2. Gas-phase chemistry, aqueous-phase chemistry, and aerosol microphysics parameterizations (gas-aerosol mass transfer, new particle formation, inter-mode transfer due to particle growth, coagulation, and aging of primary carbon particles to accumulation mode) are calculated for a 30-minute time window. Elevated emissions of aerosols and precursor gases are also applied, as well as wet removal of the gases. Rates of conversion between aerosol and gas species and between different aerosol modes are calculated, and the corresponding mixing ratio tendencies are used to update the aerosol and gas mixing ratios.

3. The surface emission fluxes of aerosols and the net fluxes (emission minus turbulent dry deposition) of precursor gases are converted to mixing ratio tendencies in the lowest model layer. These tendencies are applied over a 30-minute timestep to update the corresponding mixing ratios.

4. Dry removal of aerosols is calculated for a 30-minute time window, and the aerosol mixing ratios are updated. The dry removal equations are numerically solved with a semi-Lagrangian scheme from Rasch and Lawrence (1998) to achieve reasonable stability and accuracy.

5. The large-scale transport scheme updates mixing ratios over two 15-minute vertical remapping timesteps, each of which consists of three 5-minute sub-cycles of horizontal advection.

6. The deep convection parameterization changes EAM's temperature, humidity, and wind profiles as well as mixing ratios of hydrometeors but not yet the aerosol mixing ratios.

7. The parameterization of turbulent mixing and activation-resuspension of aerosol particles is calculated. The cloud-borne and interstitial aerosol mass and number mixing ratios are updated within this parameterization. The tendencies of cloud droplet number mixing ratio are passed on to the stratiform cloud microphysics parameterization. In default 1° EAM simulations, the turbulent mixing and activation-resuspension of aerosol particles, ice nucleation, the stratiform cloud microphysics, and the turbulence and shallow convection parameterization CLUBB are sub-cycled together using 5-minute timesteps (see Sect. 2.1 in Wan et al., 2021 and Sect. 2 in Santos et al., 2021). Within each sub-cycle, CLUBB handles the turbulent transport of heat, water, and precursor gases; CLUBB also provides the eddy diffusion coefficient, turbulent kinetic energy, and cloud fraction to the aerosol mixing and activation-resuspension parameterization. The equation of turbulent mixing of aerosols is solved not by CLUBB but in conjunction with aerosol activation-resuspension, using an explicit time-stepping method with dynamically determined stepsizes.

8. After the turbulence and cloud microphysics sub-cycles, the processes of aerosol water uptake, aerosol in-cloud

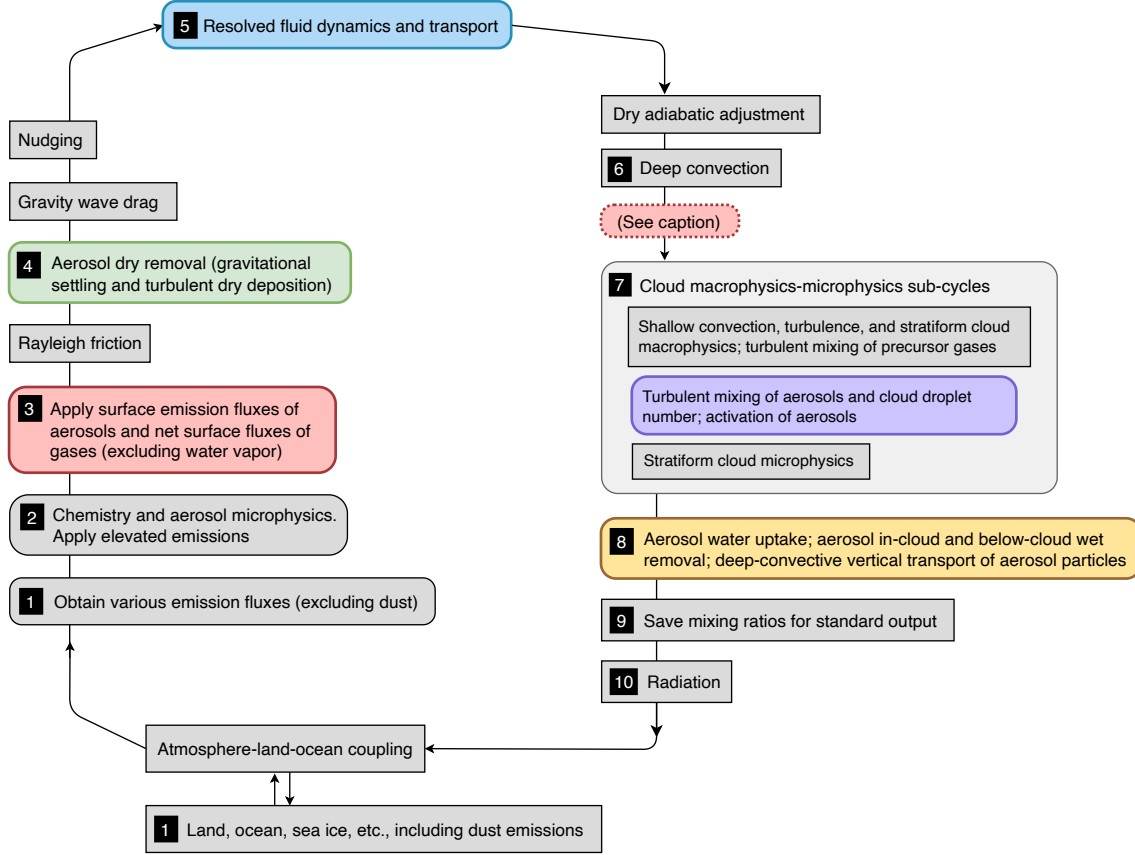

**Figure 1.** A schematic showing the sequence of calculations in a 30-minute timestep in $1°$ EAMv1 simulations. Rectangles are calculations that do not directly affect aerosol mixing ratios. Boxes with round corners are calculations that change aerosol mixing ratios. Colored boxes correspond to the physical processes for which numerical results are shown in this paper. The numbers in black squares correspond to the numbering in Sect. 2.3. The dashed box between deep convection (box 6) and the cloud sub-cycles (box 7) is where box 3 in the original EAMv1 is moved to when the revised coupling scheme is used.

and below-cloud wet removal, and the vertical transport of aerosols by deep convection are calculated for a time window of 30 minutes, and the corresponding mixing ratios are updated.

9. The mixing ratios, together with many other physical quantities describing the simulated atmospheric state, are recorded and passed to the software infrastructure that handles model output. In other words, the mixing ratios in EAM's output files that the model developers and users typically analyze are from this location in the time loop. (The mixing ratios at other locations presented later in the paper were obtained using the online diagnostic tool CondiDiag (Wan et al., 2022).)

10. The mixing ratios and other properties of aerosol particles are passed to the radiation parameterization where the aerosol impact on the atmospheric energy budget is calculated. Radiation does not directly affect aerosol mixing ratios.

After radiation and some additional diagnostics, the atmosphere model exchanges information with the other components of E3SM, and the calculation goes back to step 1 listed above.

The sequence of calculations described above is used by the original EAMv1. In the revised process coupling scheme discussed in Sects. 3.3 and 4, the surface emissions of aerosols as well as the net surface fluxes (i.e., emissions minus turbulent dry deposition) of precursor gases are applied after deep convection and before the cloud macrophysics-microphysics sub-cycles. That is, box 3 in Fig. 1 is moved to the location indicated by the dashed box between 6 and 7 when the revised coupling scheme is used.

## 2.4   Comparison with some other models

We now briefly compare EAMv1 with some other models for their assumptions about aerosol emissions at or near the Earth's surface as well as the numerical coupling of emissions, dry removal and turbulent mixing. Admittedly, it is not clear to us what the most common practices are in this re-

spect, as most of the model description papers we have read so far did not explicitly describe the discrete numerical algorithms used in coupling these processes. Here, we reference a small number of examples in which such information has been provided.

Recall that in the default EAMv1, surface emissions of aerosols are applied as a separate physical process. Within a 30 min time window in a simulation with 1° horizontal grid spacing, the direct effect of such emissions is limited to the lowest model layer regardless of layer thickness. Surface emissions, dry removal, and turbulent mixing are sequentially split in the stated order.

The assumption of non-zero dust emissions only at the Earth's surface is used in various other global models (e.g., Gong et al., 2003; Stier et al., 2005; Mann et al., 2010; Zhang et al., 2010). In the CAM3-LIAM and GAMIL-LIAM models discussed in Zhang et al. (2010), surface emissions are also a separate process directly affecting only the lowest model layer, but the sequential splitting uses the order of emissions, turbulent mixing, and then dry removal. In the version of GISS ModelE described in Koch et al. (2006) and in the IFS-AER model cycle 45R1 (Rémy et al., 2019), the surface emissions (of at least some species) and the surface dry deposition fluxes are used as boundary conditions for turbulent mixing (see Sect. 2 in Koch et al., 2006 and Sect. 2.1 in Rémy et al., 2019). In the Oslo CTM3 model described in Søvde et al. (2012), emissions, turbulent mixing, (chemistry,) and dry removal are calculated sequentially and sub-cycled together with respect to large-scale transport (see the "EBCD-sequence" described in Sect. 2 of Søvde et al., 2012).

Some models make explicit assumptions about altitude ranges that dust and other emissions can directly affect. For example, the global CTM named IMPACT (see, e.g., Liu et al., 2005; Wang et al., 2009) injects dust and some other aerosols uniformly in the boundary layer. Two additional examples in this category are the EC-Earth model configuration described in van Noije et al. (2014) and the EC-Earth3-AerChem model described in van Noije et al. (2021). There, emissions of the "surface" type (including oceanic emissions) are vertically distributed to the altitude range of 0 to 30 m. Dust emissions, on the other hand, are assigned to the "near-surface" type, for which 80% of the emissions are distributed to the 0–30 m altitude range and the other 20% to the 30–100 m range (see Table A1 in van Noije et al., 2014). The CTM named TM5, which is used in the two EC-Earth model versions mentioned above and described in Huijnen et al. (2010), employs a process coupling scheme that consists two sub-cycles during a base timestep: The first sub-cycle calculates vertical mixing by turbulence (and deep convection), then emissions with prescribed heights, and afterwards dry deposition as part of the chemistry solver; in the second sub-cycle, the sequence of calculations is reversed (Huijnen et al., 2010, Sect. 2.3).

In the CTM named EMEP MSC-W documented in Simpson et al. (2012), one can find an example where the implementation is to technically apply "surface" or "near-surface" emissions only to the lowest model layer but the effect is mixing through a substantial altitude range. In Sect. S4.5 of the supplementary material therein, it is stated that the sea salt aerosols generated by the emission parameterization are assumed to be instantaneously mixed within the model lowest layer (approximately 90 m height) at each time step.

## 2.5 EAMv1 simulations in this paper

To analyze features of the simulated dust life cycle in the 1° configuration of EAMv1, we present atmospheric simulations from October 1999 to December 2009, with the first 3 months discarded as spin-up. The specifications of external forcing, e.g., sea surface temperature and sea ice extent as well as the emissions of anthropogenic aerosols and their precursors, followed the Coupled Model Intercomparison Project Phase 6 (CMIP6) protocol of the twentieth-century transient simulations.

A total of 4 simulations were conducted, two of which used EAMv1's original process coupling scheme described in Sect. 2.3, and the other two used the revised process coupling described in Sect. 3.3. For each of the coupling schemes, we conducted one simulation with EAMv1's default L72 vertical grid and another simulation with the L30 grid following the earlier studies of Feng et al. (2022) and Zhang et al. (2018). A comparison between the near-surface layers in the L30 and L72 grids is shown in the left panel of Fig. 2.

The simulations used EAMv1's default timestep settings documented in Wan et al. (2021) and Santos et al. (2021). During a simulation, the evolution of dust mixing ratios within 30-minute time windows and the tendencies associated with boxes 3, 4, 5, 7, and 8 in Fig. 1 were tracked with the online diagnostic tool of Wan et al. (2022).

## 3 Motivation for a revised coupling scheme

This section presents budget analyses for the total mass mixing ratio of interstitial dust particles (Sect. 3.1) and discusses the weaknesses of the original coupling scheme (Sect. 3.2). A revised coupling scheme is described in Sect. 3.3 and evaluated in the next section.

## 3.1 Dust mass budget

Geographical distributions of 10-year mean dust mass emission fluxes and interstitial dust burden from the default EAMv1 are presented in panels (a) and (b) of Fig. 3. The mass burden shown here is the total mass mixing ratio summed over MAM4's accumulation mode and coarse mode, multiplied by air density, and vertically integrated over the atmosphere column. As expected, the dust emissions are

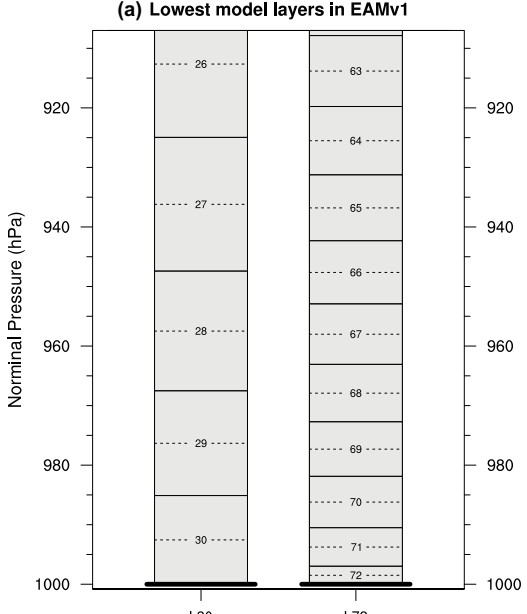

**Figure 2.** (a) Near-surface layers of the L30 and L72 vertical grids used by EAMv1. Each gray box with a solid outline and a number at the center represents one model layer. The numbers are layer indices. The dashed lines indicate locations of layer midpoints. The thick horizontal lines at 1000 hPa nominal pressure represent the Earth's surface. Here, nominal pressure refers to the air pressure at a layer interface or midpoint when the air pressure at the Earth's surface is 1000 hPa. (b) A schematic depicting the impact of layer thickness on the dust mixing ratio in the lowest model layer after surface emissions are applied. In a numerical model where the surface emissions are treated as a separate physical process and coupled with other processes using sequential splitting, given the same emission flux and same timestep size, a thinner bottom layer means a higher layer-mean mixing ratio will result after the surface emissions are applied. A discussion of this schematic can be found in Sect. 3.2.

highly inhomogeneous in space while the distribution of the dust burden is much smoother due to transport by winds.

To obtain an overview of the balance between different physical processes, we selected the box with solid outline in Fig. 3a to cover the major dust sources in Asia, Europe, and North Africa. Within the box, the grid columns with non-zero 1-year mean emissions were identified as source regions in the year, and the grid columns with no emissions were identified as source-vicinity regions in the year. The dashed box in Fig. 3a was selected as an example of a region far away from dust emissions (i.e., a remote region). Vertical profiles of 1-year mean dust mass mixing ratio tendencies were averaged over the source, vicinity, or remote regions of each year from 2000 to 2009. After that, the mean and standard deviation of the 1-year-mean regional averages were calculated to create Fig. 3c-h, where the 10-year averages are shown as markers, and two standard deviations of the yearly averages are indicated by the horizontal extent of the attached black lines.

Distinct characteristic magnitudes are seen in Fig. 3c-h in different regions and altitude ranges. The second row in the figure shows the results in the lower troposphere with the lowest model layer *excluded*, while the third row shows the 10 model layers closest to the Earth's surface, *including* the bottom layer. Furthermore, the tendencies in panels (c)-(e) and (g)-(h) have been multiplied by factors of 100 to 100,000 in order to be plotted with the same x-axis range as in panel (f). These factors of multiplication are noted at the top of each panel. A key feature revealed by these vertical profiles is that the annual and regional mean tendencies in the lowest model layer in the dust source regions are 2 orders of magnitude stronger than the tendencies in the upper layers within the same regions. Furthermore, the tendencies in the lowest model layer in the source regions are 2 to 5 orders of magnitude stronger than the tendencies in any layer in the source-vicinity or remote regions, and the characteristic magnitude differences in the 10-year averages are much larger than the interannual variabilities. The dominant source and sink terms in the global-scale annual mean dust mass budget are the following processes in the lowest model layer in the dust source regions: (1) surface emissions, (2) dry removal, and (3) turbulent mixing and aerosol activation-resuspension. (The middle row of plots in the figure reveals some discontinuities in the second or third lowest model layers. These discontinuities are commented on in Sect. 4.1).

To demonstrate that similar contrasts in the magnitudes of process rates can be seen in individual grid columns and at individual timesteps, Fig. 4 and Fig. 5 present results de-

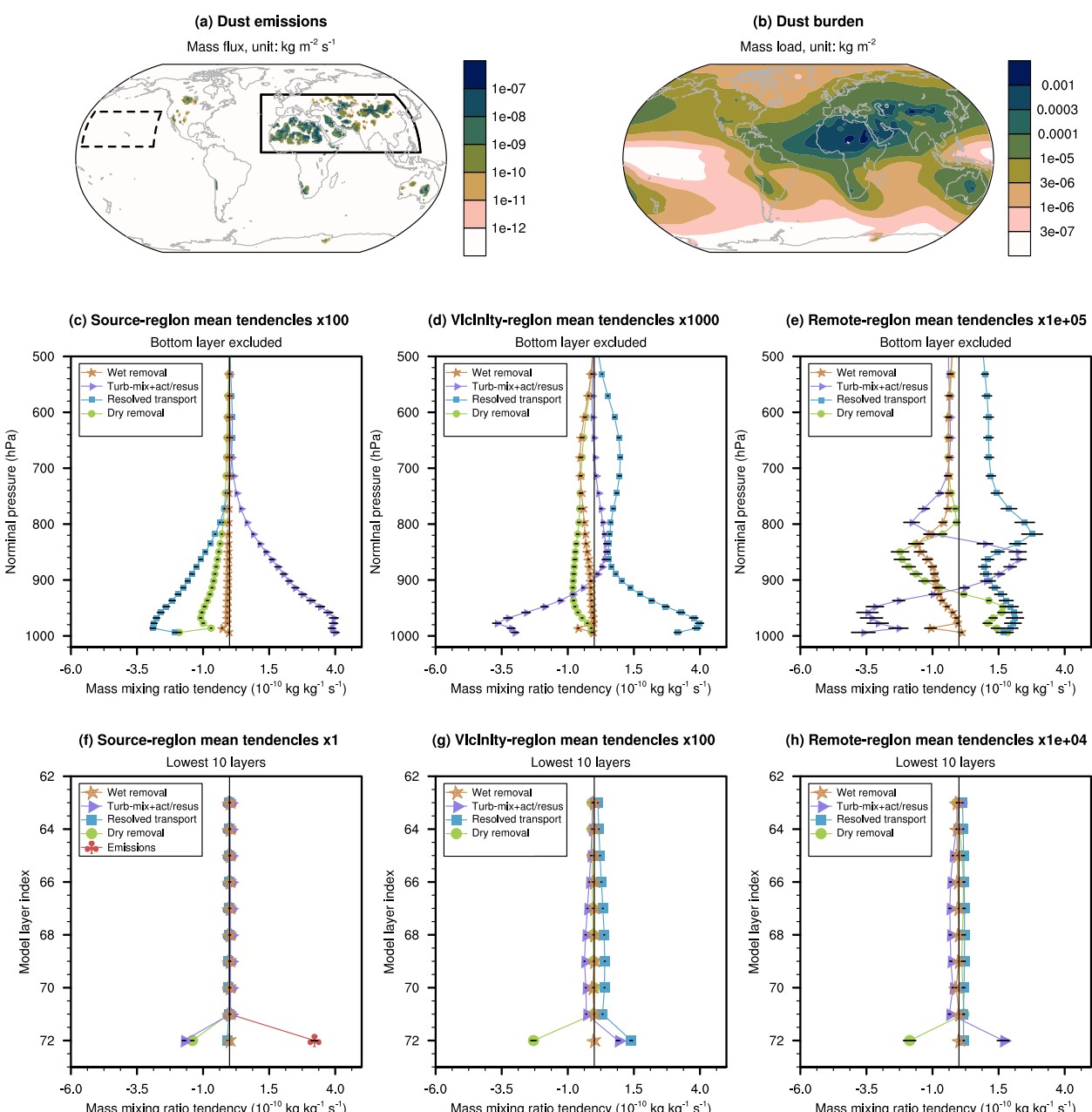

**Figure 3.** Ten-year mean annual averages of various quantities in the EAMv1 simulation conducted with the original coupling scheme and the L72 vertical grid. The first row shows the surface emission flux (panel a) and burden (panel b) of interstitial dust aerosol mass. The second and third rows are vertical profiles of the tendencies (i.e., rates of change) of interstitial dust mass mixing ratio caused by various physical processes. The left, middle, and right columns in these two rows are the averages over dust source regions, source-vicinity regions, and remote regions, respectively. The source regions and source-vicinity regions are defined as grid columns in the box with solid outline in panel (a) in which the mean emission fluxes are non-zero and zero, respectively. Remote regions refer to grid columns inside the box with dashed outline in panel (a). Panels in the second row show results of the lower troposphere, with the bottom layer of the L72 grid excluded. Panels in the last row show results in the lowest 10 model layers including the bottom layer. Emission-induced tendencies are non-zero only in the lowest model layer in source regions, and hence are shown only in panel (f). The markers of various shapes indicate 10-year averages; the horizontal line attached to each marker indicates two standard deviations of 1-year averages. Note that the tendencies shown in panels (c)-(e) and (g)-(h) have been multiplied by factors of 100 to 100,000, as noted at the top of each panel, in order for the same x-axis range to be used for all profile plots.

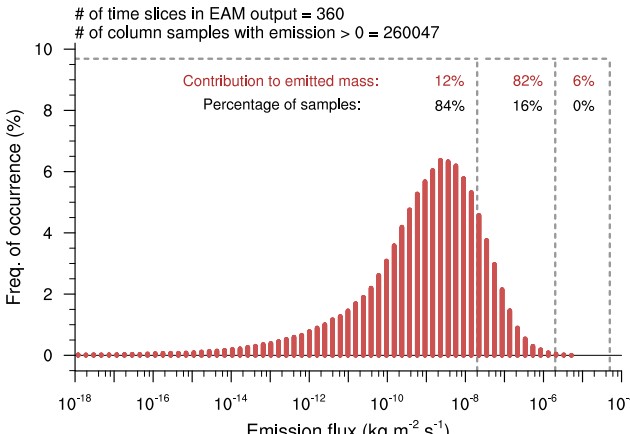

**Figure 4.** Histogram of dust emission fluxes derived from 6-hourly instantaneous model output in the 90-day period of 2000-01-29 to 2000-04-28 in grid columns with non-zero instantaneous emissions in the box with solid outline in Fig. 3a.

rived from 6-hourly instantaneous model output in the 90-day period of 2000-01-29 to 2000-04-28. In this time period and inside the box with solid outline in Fig. 3a, different grid columns and output timesteps were considered as different column samples. Figure 4 is a histogram of the dust mass emission fluxes in the L72 simulation conducted with the default EAMv1, derived from the column samples with non-zero dust emissions. The instantaneous emission fluxes span more than 10 orders of magnitude. About 82% of the emitted mass is attributable to events with mass fluxes between $2\times10^{-8}$ kg m$^{-2}$ s$^{-1}$ and $2\times10^{-6}$ kg m$^{-2}$ s$^{-1}$; about 12% of the emitted mass is contributed by a large number of weak emission events and the remaining 6% by a very small number of very strong events (Fig. 4). Given the wide range of emission fluxes seen in the histogram, we speculated and then confirmed that the column samples with very week emissions (i.e., near the left end of the histogram) exhibited properties of dust mixing ratios and process rates that were similar to the samples in the source-vicinity regions, and those properties changed gradually as the collection of column samples was shifted from the left to the right of the histogram. In order to show results from source regions that were both representative and impactful in terms of their contribution to the total emission, the middle portion of the histogram was selected for the analysis shown in the next figure.

For the middle portion of the histogram, the upper row of Fig. 5 shows statistical distributions of the instantaneous dust mixing ratio tendencies in the lowest 10 model layers of the L72 grid. The different columns correspond to different physical processes. The results for the source-vicinity and remote regions are shown in Figs. S1 and S2 in the supplementary material, respectively. These figures suggest the dominant sources and sinks of dust mass on a timestep-by-timestep and grid-column-by-grid-column basis are the same

as what we saw in the regional and annual averages, namely the following processes in the lowest model layer in the dust source regions: (1) surface emissions, (2) dry removal, and (3) turbulent mixing and aerosol activation-resuspension.

We note that the particular choice of 90-day period presented above was based on technical convenience, and the emission flux thresholds used for dividing the histogram into 3 portions were somewhat arbitrary. The analyses shown in Figs. 4 and 5 have been repeated for other times of the year and for shorter time periods like 1-month, 1-day, or 1 timestep. Although the instantaneous emission fluxes changed from time period to time period, large contrasts were robustly seen in the magnitudes of process rates between the source, vicinity and remote regions and in different altitude ranges.

## 3.2  Weaknesses of the original coupling scheme

Since sequential splitting is used in the default EAMv1 for the various aerosol processes discussed above, and the ordering is emissions, dry removal, resolved transport, turbulent-mixing-activation-resuspension, and wet removal, we show in the lower row of Fig. 5 the statistical distributions of instantaneous dust mixing ratios after each of these processes has been calculated and the tendencies have been applied (i.e., the mixing ratios have been updated). Given the nature of the sequential splitting method and the magnitudes of tendencies seen in the upper row, it is understandable that large mixing ratio spikes are seen in the lowest model layer after the surface emissions are applied (Fig. 5f). The mixing ratio profiles in Fig. 5 also indicate that although dry removal is a strong sink in the lowest model layer and resolved transport can be a significant sink, neither is sufficiently strong to remove the near-surface mixing ratio spikes. In contrast, the turbulent mixing and aerosol activation–resuspension processes are very effective in vertically smoothing the profiles. Since the dust source regions are typically dry and the lowest model layer is thin (about 20 m on average), we do not expect aerosol activation to occur there frequently; hence the smoothing effect is most likely attributable to turbulent mixing. Using the online diagnostic tool of Wan et al. (2022), we also analyzed the dust mixing ratios after each of the 5-minute sub-cycles used for the parameterizations of turbulent mixing and aerosol activation-resuspension. We found that the spikes of dust mixing ratio were typically eliminated after 1 or 2 sub-cycles (i.e., 5 to 10 minutes). This is consistent with the results from Wang et al. (2011), who showed that particles injected from ships traveling below marine stratocumulus can be lofted and mixed through the cloud-topped boundary layer within minutes (see Fig. 1c and Sect. 3.1 in Wang et al., 2011).

While the sub-timestep evolution of dust mixing ratio shown in Fig. 5 is to be expected from the use of sequential splitting and the characteristic magnitudes of process rates in the default EAMv1, the figure provides a clue that the or-

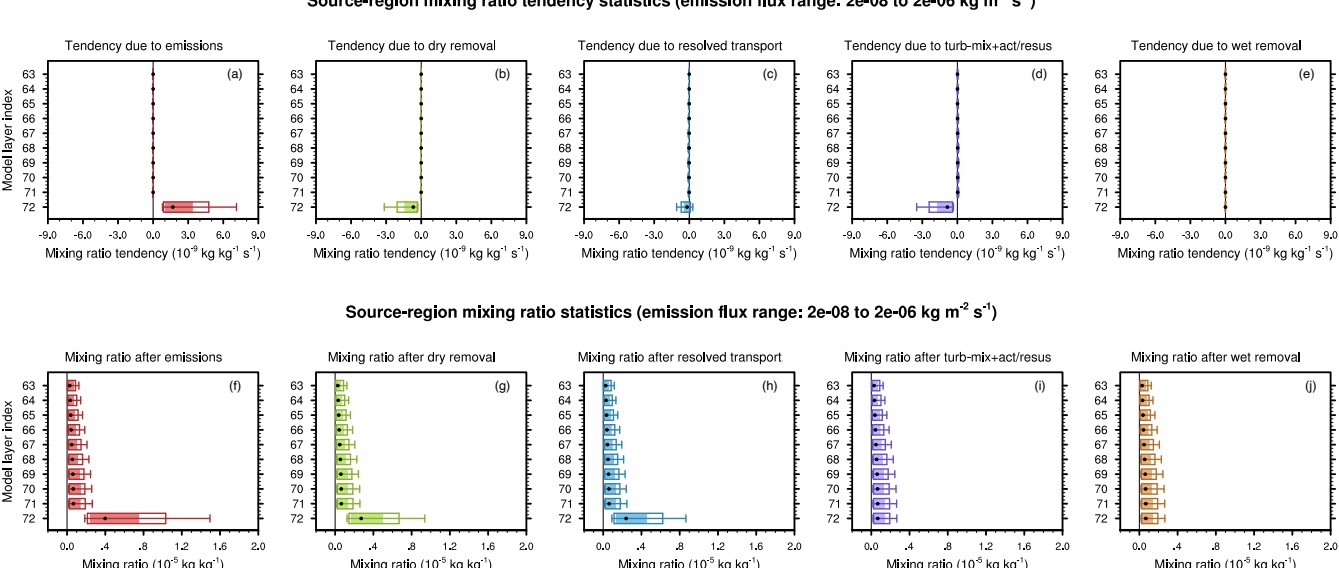

**Figure 5.** Statistical distributions of instantaneous model output in dust source regions corresponding to the middle portion of the emission flux histogram in Fig. 4. Upper row: tendencies of interstitial dust mass mixing ratio caused by various physical processes. Lower row: mixing ratio values after the corresponding tendencies in the upper row have been applied. Results are shown for the EAMv1 simulation conducted with the original process coupling scheme and for the lowest 10 layers of the L72 grid. In each panel and for each model layer, the filled box depicts the middle 50% of the statistical distribution; the outer box corresponds to the middle 67% of the distribution; the vertical whiskers are the 10th and 90th percentiles; the black dot indicates the median.

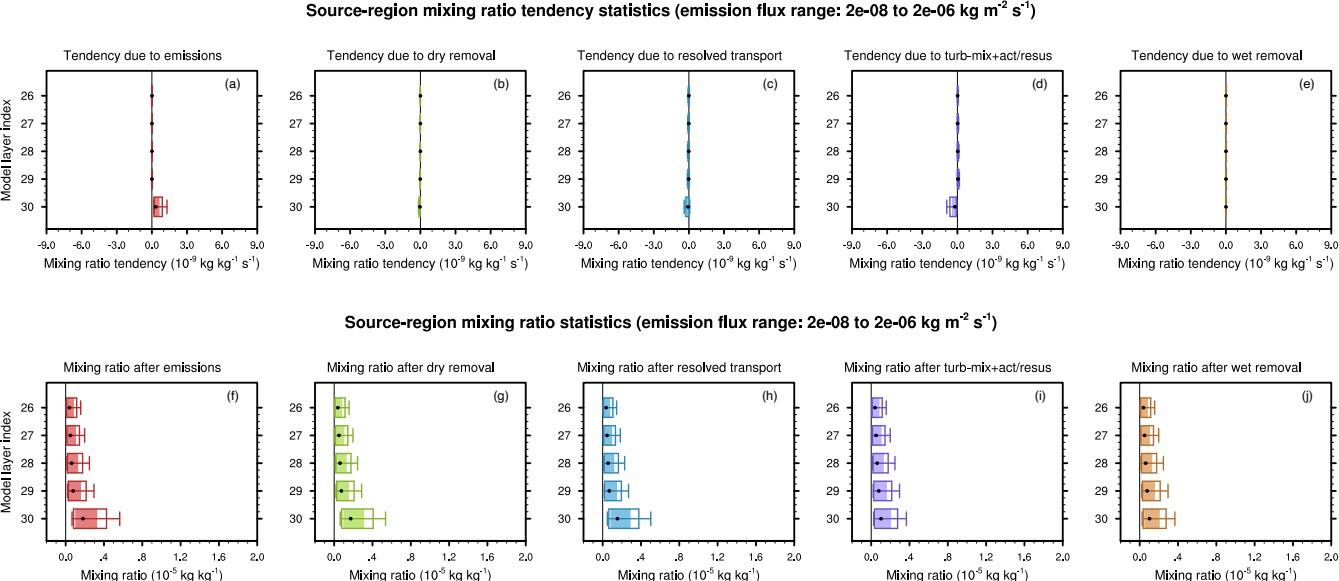

**Figure 6.** As in Fig. 5 but showing results from the EAMv1 simulation conducted with the L30 (instead of L72) vertical grid and the original process coupling scheme. Only 5 model layers are shown in this figure, as Fig. 2a suggests that the 5 lowest layers in the L30 grid cover roughly the same altitude range as the 10 lowest layers in the L72 grid.

dering of the various process has severe deficiencies. In the real world, dust emissions typically occur in turbulent atmospheric environments where the turbulent eddies are efficient in transporting the emitted particles aloft. In EAM, when the emission fluxes are applied to update the dust mixing ratios in the lowest model layer, the effect can be interpreted as immediately mixing the emitted particles throughout the lowest model layer but also temporarily trapping the particles within that layer, resulting in high concentrations being passed to the next process in the time loop, which is dry removal in the default EAMv1. Since the dry removal fluxes at the Earth's surface are proportional to the mixing ratios in the lowest model layer (see Sect. 2.2.2, Eq. 1), the high mixing ratios depicted in Fig. 5f are expected to lead to strong dry removal fluxes. If one chose, instead, to calculate turbulent mixing immediately after the emissions were applied, and then calculate dry removal afterwards (i.e., after turbulent mixing), then the input to the dry removal parameterization would look similar to Fig. 5i (i.e., without a spike in the bottom layer), hence resulting in much weaker dry removal fluxes at the surface. Following this logic, we expect the EAM simulations to be sensitive to the ordering of the emission, turbulent mixing, and dry removal processes.

The reasoning above also provides an explanation for the strong vertical resolution sensitivity of dust dry removal reported in Feng et al. (2022). As mentioned above, dust is emitted only into the lowest model layer in EAMv1, meaning the particles are temporarily trapped below the upper interface of the layer. The lowest layer in the L72 grid is about 1/5 in thickness compared to the lowest layer in the L30 grid (Fig. 2a). Therefore, given the same emission fluxes, the temporary increases in dust mixing ratio after the emissions are applied in a simulation using the L72 grid are expected to be 5 times as high as in another simulation that uses the L30 grid (see schematic in Fig. 2b), which in turn can lead to significantly stronger dry deposition in the L72 simulation. This expectation is confirmed by Fig. 6 which shows results from the simulation that used the original coupling scheme but the L30 grid. In the L30 simulation, both the tendencies and the mixing ratios spikes in the lowest model layer are substantially weaker than in L72.

It is worth clarifying that the process coupling issue identified here is not specific to EAMv1. In the predecessor model CAM5, although the parameterization of atmospheric turbulence used the scheme of Park and Bretherton (2009) and was calculated before aerosol dry removal and after surface emissions, the aerosol tracers were *not* mixed by the Park and Bretherton (2009) parameterization, but rather by the same turbulent mixing and aerosol activation–resuspension parameterization as in EAMv1. In other words, as far as the aerosol tracers are concerned, the sequence of calculation in CAM5 was the same as in EAMv1. We expect that if CAM5 simulations are performed with EAM's L72 grid, significantly stronger dry removal and shorter dust lifetime will result as well.

Furthermore, we note that the same process coupling issue also exists for other aerosol species in EAM that have surface emissions, although the magnitude of the impact depends on the relative importance of the surface emissions as well as the typical sizes of the particles. Results on this point are included in the supplementary material and briefly summarized in the conclusions.

The precursor gases in EAMv1, on the other hand, do not suffer from this coupling issue, because the splitting and ordering of the gas-related processes are different. Precursor gases in the model are assumed to experience turbulent dry deposition but no gravitational settling. The surface dry removal fluxes of gases are calculated after gas-phase chemistry inside box 2 in the schematic in Fig. 1 and then, instead of being used to update gas mixing ratios, the dry removal fluxes are subtracted from the surface emission fluxes, and the residual (i.e., the net flux) is used to update gas mixing ratios in the lowest model layer in box 3 in the schematic. Turbulent mixing of precursor gases is handled by CLUBB in box 7 in the schematic in Fig. 1.

### 3.3  A simple revision to the original coupling scheme

The weaknesses of the original coupling scheme discussed above are expected to have smaller impacts on a simulation if the overall model timestep is reduced or if the related aerosol processes are sub-cycled together so that the processes exchange information at shorter time intervals. However, both approaches would result in significantly higher computational costs, which motivates an alternate numerical scheme that couples the surface emissions of dust (as well as other aerosol species) more tightly with the turbulent mixing without changing timestep lengths. From a mathematical perspective, the ideal approach would be to provide the surface emission fluxes as a boundary condition to the equations of turbulent mixing so that the two processes (emissions and mixing) can be solved together numerically, as is done in, e.g., GISS ModelE (Koch et al., 2006) and IFS-AER (Rémy et al., 2019). This approach would require a significant amount of code modifications and is deferred to future work. Here we take a simpler and admittedly less optimal method that requires the least amount of code modifications, namely to move the update of aerosol and precursor gas mixing ratios in box 3 from the original location to the dashed box in Fig. 1, before the cloud macrophysics-microphysics sub-cycles. The emission fluxes, all other parameterizations, and the resolved dynamics are calculated at their original locations in the time loop. For aerosols, this simple modification still uses sequential splitting between emissions and turbulence-mixing-activation-resuspension, but the dry removal processes are calculated before the surface emissions are applied, and the wet removal processes are calculated after turbulent mixing; hence neither the dry removal nor the wet removal uses mixing ratios with spikes in the lowest model layer. Considering the cyclic nature of sequential split-

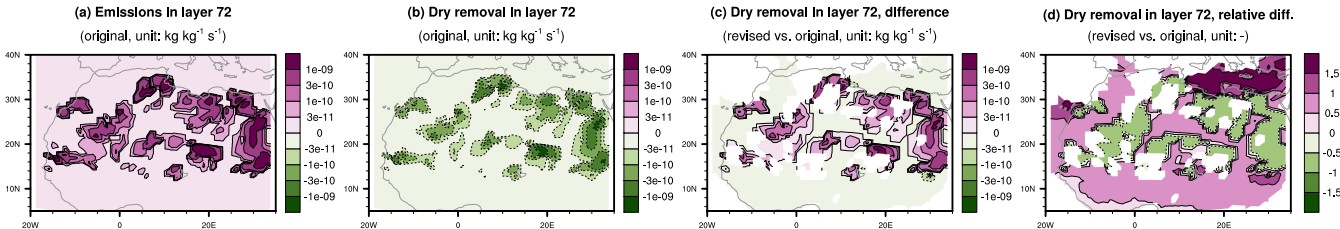

**Figure 7.** (a) and (b): 10-year mean annual mean interstitial dust mass mixing ratio tendencies (unit: kg kg$^{-1}$ s$^{-1}$) caused by emissions and dry removal in the lowest model layer in the simulation conducted with the original coupling scheme and the L72 grid. (c) and (d): The differences (unit: kg kg$^{-1}$ s$^{-1}$) and relative differences (unitless), respectively, in the 10-year mean dry removal tendency between the simulations conducted with the revised and original process coupling schemes. In panels (c) and (d), the locations masked out in white are where the differences between the two ensembles of one-year averages are statistically insignificant according to the Kolmogorov-Smirnov two-sample test with a significance level of 0.05.

ting and ignoring the other processes, this revised scheme makes the calculations in EAMv1 similar to the numerical coupling used in the CAM3-LIAM and GAMIL-LIAM models described in Zhang et al. (2010) and the "EBCD-sequence" in the Oslo CTM3 model described in Søvde et al. (2012).

## 4   Impact of the revised process coupling on aerosol climatology in EAMv1

We now compare EAMv1 simulations conducted with the original and revised coupling schemes. The goal is twofold: (1) to verify the reasoning in the previous section about the features of the two schemes, and (2) to evaluate the impacts of the revised coupling on the simulated aerosol climatology at regional and global scales. The analysis in this section focuses on 10-year mean annual averages and takes into account the interannual variability. Whether a grid column was a dust source region or non-source in a particular year and simulation was determined using the annual mean emission flux of the respective year and simulation.

### 4.1   Dust life cycle

From the discussions in Sect. 3, we expect the direct effect of the revised coupling scheme to be a weakening of dry removal in grid cells and timesteps where dust emissions occur. Based on the understanding of process interactions in EAM, we can reason how the other aerosol processes may be affected. In the revised scheme, turbulent mixing is the first aerosol process calculated after surface emissions are applied. Since the newly emitted particles have not gone through dry removal, more (compared to the case in the original scheme) particles are available for upward transport by turbulence. After turbulent mixing, more aerosol particles can be expected in upper model layers in the source regions. These particles can go through wet removal, or get advected out of the atmosphere column by resolved winds. More transport from source to non-source regions can increase aerosol

load in the non-source regions and consequently lead to more removal there. These expectations are confirmed by the EAM results shown below.

The weakening of dry removal in the lowest model layer in dust source regions can be seen in Fig. 7 which shows the L72 results in North Africa as an example. The emission-induced dust mixing ratio tendencies are shown in panel (a) to help identify locations with emissions. The changes in dry removal tendencies caused by the revision of coupling exhibit spatial patterns that closely match the emissions (Fig. 7c vs. 7a). Since dust emissions and mixing ratios at specific locations are known to have strong natural variabilities, we applied the Kolmogorov-Smirnov two-sample test at each geographical location shown on the map to compare the two 10-member ensembles of 1-year mean dry removal tendencies. The results in panels (c) and (d) are masked out in white where the probability of the two ensembles coming from the same statistical distribution is larger than 5%. The relative differences shown in Fig. 7d suggest that decreases between 50% and 100% in the 10-year mean dry removal rate can be found in the majority of the North African dust source regions.

The process rate changes in dust source regions in model layers above the lowest can be seen from the vertical profiles shown in Fig. 8. The figure is essentially the same as Fig. 3c but with the results obtained using the revised coupling added as unfilled markers. Like in Fig. 3c, the tendencies shown here were derived from annual mean values averaged over dust source regions located in the box of solid outline in Fig. 3a. The markers show 10-year averages and the attached horizontal lines indicate two standard deviations of the yearly averages. The comparison in Fig. 8 shows that when the revised coupling is used, large and systematic increases in magnitude are seen in turbulent mixing, resolved transport, and dry removal in the lower troposphere above the lowest model layer. Between 900 hPa and 1000 hPa, the relative magnitude increases are on the order of 100%. The changes in the 10-year mean values are large compared to the interannual variabilities.

**Table 1.** Impact of the revised process coupling on the life cycle of interstitial dust aerosol in EAMv1 simulations conducted with the L72 vertical grid for the years 2000-2009. Shown here are the 10-year mean, vertically integrated sources, sinks, and burden of interstitial dust aerosol mass summed over all grid columns with dust emissions, all grid columns without dust emissions, or the entire globe. The lifetime values shown in the last row were calculated from the corresponding values of total burden and total source. Units of the process rates, mass burden, and lifetime are indicated in the left most column. The percentages shown in parentheses are the standard deviation of 1-year averages normalized by the 10-year mean. The percentages shown without parentheses are the relative differences in 10-year mean between the simulation using the revised coupling scheme and the simulation using the original scheme.

| Dust budget | Source-region total | | | Non-source-region total | | | Global total | | |
|---|---|---|---|---|---|---|---|---|---|
| | Original | Revised | Rel. diff. | Original | Revised | Rel. diff. | Original | Revised | Rel. diff. |
| Emissions [Tg yr$^{-1}$] | +4500 ($\pm$3%) | +4603 ($\pm$3%) | +2% | - | - | - | +4500 ($\pm$3%) | +4603 ($\pm$3%) | +2% |
| Dry removal [Tg yr$^{-1}$] | $-$2688 ($\pm$4%) | $-$1723 ($\pm$4%) | $-$36% | $-$776 ($\pm$5%) | $-$1396 ($\pm$4%) | +80% | $-$3464 ($\pm$4%) | $-$3118 ($\pm$4%) | $-$10% |
| Activation [Tg yr$^{-1}$] | $-$102 ($\pm$9%) | $-$151 ($\pm$10%) | +48% | $-$324 ($\pm$4%) | $-$450 ($\pm$7%) | +39% | $-$427 ($\pm$4%) | $-$601 ($\pm$7%) | +41% |
| Wet removal [Tg yr$^{-1}$] | $-$112 ($\pm$5%) | $-$196 ($\pm$7%) | +75% | $-$496 ($\pm$6%) | $-$687 ($\pm$3%) | +38% | $-$608 ($\pm$5%) | $-$883 ($\pm$3%) | +45% |
| Resolved transport [Tg yr$^{-1}$] | $-$1597 ($\pm$4%) | $-$2533 ($\pm$4%) | +59% | +1597 ($\pm$4%) | +2533 ($\pm$4%) | +59% | - | - | - |
| Burden [Tg] | +7.6 ($\pm$4%) | +11.4 ($\pm$4%) | +49% | +14.8 ($\pm$6%) | +20.0 ($\pm$4%) | +35% | +22.5 ($\pm$5%) | +31.4 ($\pm$4%) | +40% |
| Lifetime [day] | +0.6 ($\pm$2%) | +0.9 ($\pm$2%) | +46% | +3.4 ($\pm$3%) | +2.9 ($\pm$3%) | $-$15% | +1.8 ($\pm$2%) | +2.5 ($\pm$3%) | +37% |

**Figure 8.** Same as Fig. 3c but comparing results obtained with the original coupling scheme (filled markers) and the revised coupling scheme (unfilled markers). The results shown were derived from 1-year mean profiles averaged over dust source regions located in the box with solid outline in Fig. 3a. The markers show 10-year averages. The horizontal extent of an attached black line indicates two standard deviations of 1-year averages.

Furthermore, in the middle row of Fig. 3, some of the profiles exhibit discontinuities in the second or third lowest model layers. These features are possibly non-local effects of the very strong discontinuities between the second lowest and lowest model layers shown in the third row of that figure and in Fig. 5. Here in Fig. 8, the results are much smoother in the lowest few layers when the revised coupling is used.

Global impacts of the revised coupling can be seen in Table 1, which shows the sources, sinks, and burden integrated over all source regions, all non-source regions, and over the globe, as well as the lifetime derived from the integrals. The dust mass sinks attributed to activation are from the parameterization of turbulent mixing and activation-resuspension, noting that the mass-weighted column integral of the mixing ratio tendency caused by turbulent mixing vanishes. For interstitial aerosols, the column-integrated wet removal rates indicate the net effect of below-cloud wet removal (which converts interstitial aerosols to precipitation-borne aerosols) and aerosol resuspension from evaporating precipitation (which converts precipitation-borne aerosols to interstitial aerosols). The numbers in the table further confirm that the revised coupling results in substantially weakened dry removal in dust source regions (a 36% decrease in the 10-year mean), stronger transport to non-source regions (a 59% increase), and an overall (global mean) weakening of dry removal as well as strengthening of wet removal and activation. The global burden of interstitial dust mass increases by 40% and the global mean lifetime increases by 37% in terms of 10-year averages. For the budget terms shown in the table, the standard deviation of 1-year averages are typically a few percent of the 10-year mean (see numbers given in parentheses in the table), hence we can conclude the changes in dust life cycle caused by the revision in coupling scheme are large and significant compared to interannual variability.

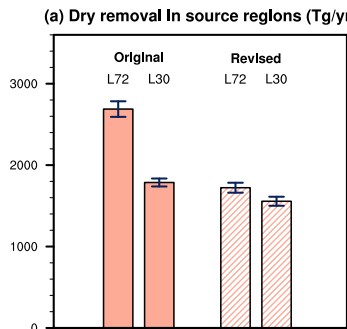
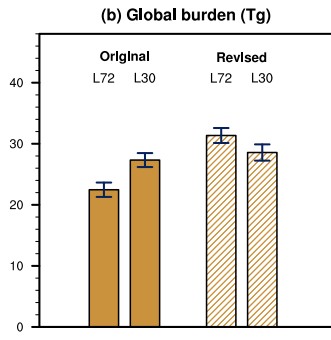
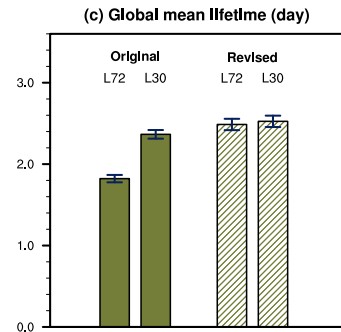

**Figure 9.** Comparison of simulations conducted with the L72 or L30 vertical grid and using the original or revised coupling scheme. (a) Dry removal rate of interstitial dust mass integrated over all dust source regions of the globe, shown here as positive values (unit: Tg yr$^{-1}$). (b) Global burden of interstitial dust mass (unit: Tg). (c) Global mean lifetime of interstitial dust mass (unit: day). The filled or hatched rectangles show 10-year averages. The whiskers indicate two standard deviations of 1-year averages.

## 4.2 Sensitivity to bottom layer thickness

In Sect. 3.2, we attributed the strong vertical resolution sensitivity of dust dry removal reported in Feng et al. (2022) to the original sequential splitting scheme that calculates dry removal after surface emissions and before turbulent mixing. The revised coupling scheme allows the emitted particles to be vertically mixed by turbulence before other processes are calculated. Since turbulent mixing is very efficient in reducing the vertical gradients and resulting in similar mixing ratios in model layers near the Earth's surface (Fig. 5i), we expect the simulated dry removal rates to be less sensitive to the thickness of the lowest model layer when the revised coupling is used. This expectation is confirmed by Fig. 9a which shows the annual mean dry removal rate integrated over all source regions on the globe. Furthermore, panels (b) and (c) in the figure suggest that for the simulated global burden and lifetime, the results are also less sensitive to bottom layer thickness when the revised coupling scheme is used.

## 5 Summary, conclusions, and outlook

The earlier work by Feng et al. (2022) pointed out that various aspects of the dust aerosol life cycle simulated by EAMv1, in particular the dry removal fluxes and lifetime, were sensitive to the use of 72 versus 30 layers for the vertical grid. In this paper, we investigated the resolution sensitivity by carrying out detailed budget analyses for the interstitial dust mass mixing ratio and by tracking the evolution of the mixing ratio within the model's time integration cycle. Dust emissions in the real world typically occur in turbulent ambient conditions, hence the emitted particles can be efficiently distributed to a significant depth of the atmosphere column. In EAMv1, the numerical coupling method treats surface emissions as a separate process which adds aerosol particles to the lowest model layer, while dry removal and turbulent mixing are calculated after the emissions are applied, using the sequential splitting method. This ordering of processes results in high, unrealistic temporary values of aerosol mixing ratios to be provided as input to the parameterization of dry removal, causing large numerical errors in the simulated dry removal rates. The problem is exacerbated when the model's bottom layer is thinner, as the same emission fluxes will result in higher temporary mixing ratios in the bottom layer when emissions are applied in isolation.

Based on this reasoning, we proposed a simple revision to the numerical process coupling in EAMv1 that required the least amount of code changes among all the alternative schemes considered, namely to move the application of emissions to the location right before turbulent mixing in the model's time integration loop. The revision allows the newly emitted aerosol particles to be vertically distributed by turbulence before experiencing other processes considered in the model, hence giving a numerical coupling scheme with closer resemblance to the process interactions in the real world.

Transient atmospheric simulations were conducted for the years from 2000 to 2009 using 72 or 30 grid layers with 1° horizontal grid spacing. As expected, the revision substantially weakened dry removal and strengthened vertical mixing of dust in its source regions, strengthened the large-scale transport from source to non-source regions, strengthened dry removal outside the source regions, and strengthened activation and wet removal globally. When using 72 grid layers without retuning uncertain parameters of emission strength, the revised process coupling was found to cause a 40% increase in the 10-year mean global dust burden and an increase of the 10-year mean global mean dust lifetime from 1.8 days to 2.5 days.

The revised process coupling was found to affect all aerosol species with substantial surface emissions, i.e., dust, sea salt, MOA, BC, and POA, leading to weaker dry removal as well as higher mixing ratios throughout the atmosphere. The resulting changes in mass mixing ratio were large for the species found mainly in large aerosol particles (i.e., dust and

sea salt) and were considerably smaller for the species found mainly in submicron particles (i.e., MOA, BC, and POA). These results can be found in the supplementary material.

Numerical experiments also confirmed that the revised coupling significantly reduced the strong and nonphysical sensitivities of model results to vertical resolution in the original EAMv1, providing a justification for adopting the revised scheme as well as a motivation for future improvements. The revision proposed here was very simple in terms of the amount of code changes it required. Like in the original scheme, the surface emissions and dry removal of aerosols were still sequentially split from turbulent mixing using relatively long coupling timesteps of 30 minutes. In the next step, we plan to numerically solve the turbulent mixing equations using surface emissions and turbulent dry deposition as boundary conditions, as has been done in Koch et al. (2006) and Rémy et al. (2019). We also plan to explore different ways to couple these processes with gravitational settling. On the one hand, we do not expect vertical resolution sensitivities in the simulated aerosol life cycles will be completely eliminated merely by further revising the numerical coupling of aerosol processes discussed in this paper, because discretization errors in the individual parameterizations, as well as discretization errors and numerical coupling errors associated with other processes in EAM (e.g., clouds), can also cause vertical resolution sensitivities, and some of the sensitivities can be advantageous as they demonstrate the benefits of using higher resolutions. On the other hand, since process coupling is one of the error sources in multi-process models and given the results shown in this paper, further work on numerical coupling will likely make useful contributions to the goal of reducing numerical errors in EAM simulations.

*Code and data availability.* The EAMv1 source code used in this study can be found on Zenodo as record 7995850 (Wan, 2023). The simulations scripts and analysis scripts can be found on Zenodo as record 10371316 (Wan and Zhang, 2023a). The EAMv1 simulation output analyzed in the paper can be found on Zenodo as record 10407375 (Wan and Zhang, 2023b).

*Author contributions.* RCE and PJR initiated this study. HWan proposed and implemented the revised process coupling. HWan and KZ conducted and analyzed the EAM simulations with input from the coauthors. HWan wrote the manuscript with input from the coauthors. All coauthors contributed to the revisions.

*Competing interests.* The authors declare that no competing interests are present.

*Acknowledgements.* The authors thank the two anonymous referees for their careful reviews and insightful questions and suggestions. This work was supported by the U.S. Department of Energy's (DOE's) Scientific Discovery through Advanced Computing (SciDAC) program via a partnership in Earth system model development between DOE's Biological and Environmental Research (BER) program and Advanced Scientific Computing Research (ASCR) program. KZ, YF, and HWang were supported by DOE BER through the E3SM project. The work used resources of the National Energy Research Scientific Computing Center (NERSC), a DOE Office of Science User Facility located at Lawrence Berkeley National Laboratory, operated under Contract No. DE-AC02-05CH11231, using NERSC awards ASCR-ERCAP0025451. Computational resources were also provided by the Compy supercomputer operated for DOE BER by the Pacific Northwest National Laboratory (PNNL). PNNL is operated for the U.S. DOE by Battelle Memorial Institute under contract DE-AC06-76RLO1830. The work by Lawrence Livermore National Laboratory was performed under the auspices of the U.S. Department of Energy under Contract DE-AC52-07NA27344. LLNL-JRNL-850087-DRAFT. YF acknowledges the support of Argonne National Laboratory (ANL) provided by the U.S. DOE Office of Science, under Contract No. DE-AC02-06CH11357.

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
