# Peer review of "Numerical coupling of aerosol emissions, dry removal, and turbulent mixing in the E3SM Atmosphere Model version 1 (EAMv1), part I: dust budget analyses and the impacts of a revised coupling scheme"

_EGUsphere, 2023_

## Author Response (AR1)

**Point-by-point Response to Referee Comments**

Dear Editor,

On behalf of the co-authors, I would like to thank the two anonymous referees for their very careful reviews and very constructive and helpful comments. In response to the comments, we have made the following major changes to the manuscript:

1. Added 2 paragraphs in the Introduction section with general comments on process coupling in aerosol-climate models as well as in the broader research area of weather, climate, and Earth system modeling. Added a new subsection (2.4) with a literature review on the specific problem of coupling aerosol emissions, turbulent mixing, and dry removal.
2. Replaced the 1-year wind-nudged simulations in the original manuscript with 10-year free-running simulations. Updated all figures and tables. Added information of interannual variability to all figures and tables where 10-year mean results are shown. (Please note that the 10-year averages shown in the revised manuscript have turned out to be very similar to the 1-year averages shown in the original manuscript, suggesting our key results are robust.)
3. Shortened the manuscript by
   a. Reducing the number of tables from 8 in the original manuscript to 1 in the main body of the revised manuscript and 1 in the supplementary material.
   b. Reducing the number of figures from 12+3 in the original manuscript to 9 in the main body and 3 in the supplementary material.
   c. Moving to the supplementary material (and reducing the contents of) the discussions on aerosol species other than dust.
   d. Removing some detailed descriptions of the EAM model that are not essential to understanding the key results of this paper.
4. Replaced colormaps in all contour plots with palettes from the Scientific Color Maps 8.0 package recommended by GMD. Revised line plots by using more distinguishing marker shapes and line styles. Removed all explicit mentions of colors in the text.
5. Posted the simulation scripts and analysis scripts on Zenodo at https://doi.org/10.5281/zenodo.10371316

We have carefully addressed all comments from the referees. The point-to-point responses can be found on the following pages, with the referees' comments shown in blue and our responses shown in black.

We believe the manuscript has been improved significantly, for which we thank the referees again for their careful reviews and valuable insights.

Sincerely,

Hui Wan

**Response to Referee 1**

Wan et al. show that in an atmosphere model at a 1° horizontal resolution, a coupling of surface emissions to the planetary boundary layer turbulence parameterization becomes even more important with increasing vertical resolution. I find this result not only plausible, but expected. What seems surprising to me is rather that in standard EAMv1, the vertical mixing of emissions prior to dry deposition was entirely determined by the thickness of the lowest model layer. As far as I know, global atmosphere models (sometimes with a vertical resolution on the order of 100m near the surface) usually assume instantaneous vertical mixing of surface emissions extending at least across the height of the lowest model layer (perhaps excluding point sources in a few models). But up to now, I also thought that many models either already couple surface fluxes and surface emissions to the planetary boundary layer turbulence parameterization (for example similar to what is mentioned for the IMPACT model on page 6) or else call the turbulence parameterization right after the emissions or surface flux computation, as suggested in the present manuscript.

The problem with the instantaneous artificial vertical mixing across the height of the lowest model layer is that, unlike parameterized mixing, it is not based on physical arguments and independent of local stability and the actual boundary layer height, which varies in space and time including a dependence on surface type, latitude, and a prominent diurnal cycle over land. Although this artificial mixing in the lowest layer may to some extent mimic boundary layer mixing in low-resolution models, I think that it is definitely not desirable. This artificial mixing can however, be reduced by increasing vertical resolution as far as numerical stability permits. A proper coupling of surface fluxes to the planetary boundary layer turbulence parameterization seems advisable in any case, and becomes even more important as the vertical resolution near the surface increases. My guess is that even at very high resolution, the process splitting discussed here may still be important as long as vertical resolution near the surface decreases proportionally.

On the one hand, a large number of well known papers on process splitting exists especially in the chemistry transport community (see minor point #1 below for an example), and perhaps elsewhere. The treatment of surface emissions, especially from point sources, has long been a concern in coarse resolution chemistry transport and aerosol models as well. For point sources, various approaches for addressing the problem of finding the right emission height and also of instantaneous mixing because of finite grid resolution have been suggested and tested, such as plume-in-grid models. Furthermore, as mentioned above, some atmosphere models may already couple emissions to the planetary boundary layer turbulence parameterization or use the process splitting suggested here.

However, based on a quick literature search, it seems to me that the publication of the manuscript by Wan et al. would be important and timely. I did not find anything concrete on this topic (although I must admit that I spent a limited time on my literature research). Because of the major effect the process splitting has on the results, the importance of the results

presented by Wan et al. for the general progress in the development of EAM seem obvious to me. I would also expect the general result to apply at least to some other models as well.

I find that the manuscript by Wan et al. is exceptionally well written, and I have only a few minor comments. While some modellers may not be overly surprised by the findings, I think that documenting this type of sensitivity and especially also suggesting a solution is extremely useful. Although I partially agree with the authors on their more cautious statements in the introduction, which suggest to me that one should think in some depth about these issues, the relative correctness of the solution suggested by the authors seems rather obvious to me.

We thank the referee for their appreciation of the value of the work. Our impression of what exists and what is hard to find in the literature on process splitting is consistent with the referee's comments. Within the atmosphere sciences, the chemistry transport modeling community has paid a substantial amount of attention to the accuracy and stability of time integration methods, including process splitting. But in global weather, climate, and Earth system models (including aerosol-climate models), when it comes to the coupling among the various parameterized physics as well as the coupling of parameterizations with the resolved fluid dynamics, detailed documentation on the splitting methods being used and concrete explanations of the reasoning behind those choices are hard to find. Like pointed out by referee 2 and by the review paper of Gross et al. (2018, MWR, doi: 10.1175/MWR-D-17-0345.1), process splitting/coupling has been a largely overlooked topic. We fully agree with referee 1 that the results presented in the manuscript are not surprising, and we expect many readers will, like the referee, find the relative correctness of the revised coupling scheme rather obvious. On the other hand, the feedback we received from a number of aerosol physicists and modelers was that why there was a numerical problem - and how to resolve it - were not obvious to them until we explained the details, and our "more cautious statements in the introduction" mentioned by the referee were expressed from these colleagues' perspective. Our hope is that publishing concrete and easy-to-understand examples like ours in peer-reviewed journals will help remind climate model developers of the potentially dominant role of splitting errors and thereby help reduce the chance of "obviously" suboptimal coupling methods being used in the future, especially when new and/or more sophisticated process representations are brought in as new modules or packages.

I would very much appreciate if the authors would consider a follow-up study to check if and how much this revision affects ERFari+aci

We have conducted pairs of simulations using the present-day and preindustrial emissions of anthropogenic aerosols and precursors, and we found the impacts on ERFari+aci to be very small. Since anthropogenic aerosols are mostly submicron species, and since our results presented in the manuscript show that the revision in process coupling has relatively small impacts on submicron species, we think the small responses in ERFari+aci are understandable.

Minor points:

1. I generally think that it is good practice to discuss the results of any given study in the context of existing literature. Therefore, at least at first, it seemed to me that the authors could have mentioned and perhaps included a brief survey of the existing literature on process (or operator) splitting in the introduction. Initially, I also thought it would be rather easy for me to suggest a few references that are of direct relevance to the issue at hand (such as the one mentioning the IMPACT model on page 6). However, after (admittedly a rather quick) literature search, I found myself mistaken. I did not find documentation of aspects that I more or less took for granted, at least not where I had expected to find it. In case the authors have more luck and/or patience, I think they could include a brief discussion of existing literature that is relevant to the topic beyond what is included in the method section the second paragraph of Section 2.2.1. My own brief literature research suggested to me that this manuscript is very timely and that it would require some effort to find studies that are directly relevant for this study should they exist. I do not think that it would be worthwhile to include a detailed discussion of other references only to show that some existing literature dealt with rather different aspects of process splitting in order to motivate this study. However, simply mentioning that some other studies have dealt with process splitting might be worthwhile. https://doi.org/10.1016/S0377-0427(99)00143-0 could perhaps serve as one starting point regarding the operator splitting literature, and there might be other good starting points. I also found https://doi.org/10.1029/2018MS001418 interesting, which the authors cited in a previous work on time step convergence, although it is not directly related to the topic of this study. Especially in case the authors do find papers which are more closely related than the ones suggested here, a brief discussion would of course be interesting.

In the revised manuscript, we have added two new paragraphs to the introduction section:

The first paragraph starts with a brief statement that the chemistry transport modeling community has accumulated rich experience in splitting methods. We then take the AeroCom Phase III model references as an example to express our wish to see more documentation and discussions on numerical process coupling in aerosol-climate models.

The second new paragraph starts with a quote from a recent (2016) book on splitting methods which states that "*practitioners of the above [splitting] methods have become quite specialized, forming subcommunities with very few interactions between them*". We then cite Gross et al. (2018) and a number of other papers to mention that process coupling is a largely overlooked topic in the broader community of weather, climate, and Earth system modeling but some recent progress has been made.

In section 2, we have added a new subsection (Sect. 2.4) that includes examples of how emissions and turbulent mixing are coupled in other aerosol-climate models or chemistry transport models. Please see more details below and in the revised manuscript.

2. 2nd paragraph of Section 2.2.1 starting from "While ...": Perhaps this could be woven into the introduction?

We have moved this to a newly added subsection (2.4 in the revised manuscript, "Comparison with some other models"). Please see also our response to the next comment.

3. e.g., Gong et al., 2003; Stier et al., 2005; Mann et al., 2010; Zhang et al., 2010 -> It would be interesting to know how process splitting was handled in these cases. In case the information is hard to find without consulting the codes, the authors could consider mentioning that they did not find the information. In the case of Zhang et al. the authors may be able to comment based on the code without first having to download it, but citing studies or documentation would be preferred.

In the newly added Sect. 2.4 titled "Comparison with some other models", we briefly compare EAMv1 with some other models for their assumptions about aerosol emissions at or near the Earth's surface as well as the numerical coupling of emissions, dry removal and turbulent mixing. We candidly admit it is not clear to us what the most common practices are in this respect, as most of the model description papers we have read so far do not explicitly describe the discrete numerical algorithms used in coupling these processes. Nevertheless, apart from explaining how two of the models in Zhang et al. (2010) handled the splitting, we have also included references to several model description papers in which the numerical coupling has been documented and/or the assumptions about emission heights have been mentioned. Please see the revised manuscript (new Sect. 2.4) for the details.

4. First line of Sect. 2.3: I am not sure simply "timestep" is the right word here. The timestep for advection is 5 minutes. A 30 minutes time step at 1° horizontal resolution seems incompatible with the Courant–Friedrichs–Lewy condition.

In the revised manuscript, we have changed "timestep" to "time window" in this sentence and at relevant later locations. Indeed, the timestep for horizontal advection handled by the dynamical core is 5 minutes. The 30-minute "timestep" is usually referred to as the "physics timestep" by the EAM developers; to be more accurate, this is the step size used by many (although not all) of the parameterizations and by the coupling among most (again not all) parameterizations. In a previous paper of ours (Wan et al., 2021, GMD, doi: 10.5194/gmd-14-1921-2021), we referred to the 30-minute timestep as the "main coupling timestep" in EAMv1. Hopefully the revised wording using "time window" is sufficient here in this manuscript.

5. I think that the long time step of 30 minutes for which dry removal is computed may affect the results. I thought that such long time steps were mainly used for radiation. Can you comment on this, perhaps somewhere in the discussion of your results? I do not expect additional sensitivity studies here. A very brief discussion in one or two sentences would suffice.

In EAM, the dry removal equations are numerically solved with a semi-Lagrangian scheme from Rasch and Lawrence (1998) to achieve reasonable stability and accuracy. We've added this note to Sect. 2.3, list item #4, in the revised manuscript.

Our study presented here focused on process coupling and did not investigate the impact of time integration methods used for individual processes. In EAMv1 (and EAMv2 as well as the candidate configuration for v3), not only dry removal, but also various other aerosol processes, including several aerosol microphysics processes and their coupling to gas-phase chemistry, are integrated using 30-minute timesteps. The accuracies of time integration in these processes and their coupling are worth evaluating in the future.

6. The decreasing sensitivity to the thickness of the lowest layer is briefly discussed in Sect. 4.3. I think that although the discussion is brief, it provides sufficient detail. In my opinion this result has important practical implications and also provides strong support and motivation for adopting the new coupling and/or for further improving on this simple approach. I would suggest to repeat this important result in Section 5 (Summary, conclusions, and outlook).

Thanks for the suggestion. We agree the reduced sensitivity to bottom layer thickness is a very important point. In the revised manuscript, we have added text on this point both in the last section and in the abstract. We also replaced some of the table contents in the original manuscript with a figure on vertical resolution sensitivity (new Fig. 9) to further highlight this point.

Other suggestions:

P. 1: to after dry removal and before turbulent mixing -> to before turbulent mixing and after dry removal (Please choose which version you find easier to read. I like the second version better because it emphasizes that in the revised scheme, the effect of turbulent mixing is computed prior to dry removal in the next time step.)

Thanks for this suggestion which reminded us of the repeatedly encountered challenge in using "before" and "after" when describing process ordering, as different modelers might take different perspectives when defining the beginning of a timestep. In the revised manuscript, we have changed the sentence to say that the revised scheme "applies emissions before turbulent mixing instead of before dry removal", in order to avoid using "after".

P. 9: I suggest to omit "stored in the Fortran variable cam_in%cflx" and also "those additional mixing ratio values were included in the output files under different variable names following the convention described in Wan et al. (2022)" on Page 10. This sounds like something that can better be included in a user manual or so.

Both have been addressed as suggested.

P. 13 To get an overview -> To obtain an overview

Changed.

P. 13 to represent regions ... ( i.e. the remote regions) -> as an example of a region ... (i.e. a remote region)

Revised as suggested.

P. 13: averaged over ... remote regions -> averaged over ... remote region

Revised.

P. 16 mush -> much

Corrected.

P. 16: I suggest to remove or replace the references to cam_in%flx, on this page and also on the following pages.

Revised as suggested.

P. 18: Are the authors aware of a model in which this approach has been taken? How did Zhang et al. (2012) handle this?

In the revised manuscript, we have added the comment that considering the cyclic nature of sequential splitting and ignoring other processes, the revised scheme makes the calculations in EAMv1 similar to the numerical coupling used in the CAM3-LIAM and GAMIL-LIAM models documented in Zhang et al. (2010) and the ``EBCD-sequence'' in the Oslo CTM3 model described in Søvde et al. (2012),

P. 18: Box 3 in Fig. 1 is moved to the location indicated by the pink box with dashed outline, after box 6 (deep convection) and before box 7 (in which turbulent mixing is calculated) -> perhaps just indicate the processes and point to Fig. 1. You mentioned the moving of the boxes in the end of Sect. 2.3, and this sentence struck me as a repetition.

Agree that this was a repetition. The sentence has been removed, and the preceding sentence has been updated to include a pointer to Fig. 1.

P. 20: The global impacts -> Global impacts

Changed.

P. 21: The first row ... <- sounds like a repetition of the figure caption. I think it can be omitted.

Thanks for the suggestion. The revised manuscript no longer has this paragraph and figure, as we reduced the contents in response to referee 2's suggestion.

P. 21: "the location marked as 3" -> I think you could omit the first two sentences of Sect. 4.2 and instead simply say that the process splitting was modified not only for dust as described in Sects. 2.3, but also for other species.

Thanks. The text has been revised as suggested. (And the original Sect. 4.2 has been moved to the supplementary file and the contents have been reduced to address referee 2's suggestion of making the paper shorter.)

P. 25: situation gets worse -> situation becomes worse Or: the problem is exacerbated

Changed to "the problem is exacerbated".

Figure 1: Perhaps add "see caption" to the box between 6 and 7. I first thought about suggesting to simplify this figure. But looking for concrete suggestions, I came to the conclusion that it is actually nice that the authors included several boxes for which the relevance is understandable on second thought.

Thanks for the thoughtful suggestion. The diagram has been updated as suggested.

Figure 9d: Wouldn't it be better to use a more linear color scale in Figure 9d?

A linear color scale is used in the revised manuscript. In order to accommodate the large values and strong spatial gradient and also to accommodate the use of a color-vision-deficiency-friendly colormap, we have chosen to use a relatively large contour interval of 0.5 (50%).

**Responses to Referee 2**

**General comments**

Wan et al. describe how a reordering of aerosol processing computations in EAMv1 reduces the sensitivity of the model's global dust cycle to vertical resolution. Due to the small vertical extent of the lowermost model layer in the 72-level resolution in connection with the isolated treatment of the surface emissions flux, intra-timestep mixing ratio values become very large, and thus the whole budget strongly depends on the process that is computed next. In the current version, this is dry deposition, which is thus strongly overestimated in source regions, at the expense of other processes, specifically long-range transport and wet removal. Moving the emissions computation directly before the turbulent transport treatment, the authors obtain a dust budget that is less sensitive to the vertical extent of the lowermost model layer, and thus less sensitive to the vertical model resolution.

I agree with the first referee that the manuscript is very well written, and also find a lack of (quickly accessible) literature on the topic of "operator splitting" in the atmospheric aerosol modeling context. It is certainly worthwhile to publish also such "overlooked" issues (as has recently also been done by Kawai et al., 10.1029/2022ms003128, for instance). In my opinion, the authors have done a very good job in thoroughly testing their suggested changes, and in documenting the tests and their results. For a publication in GMD, however, I recommend one of the following revisions.

When focusing purely on the vertical resolution (or lowermost layer thickness) sensitivity, i.e., on a rather technical problem and its solution, I suggest to considerably shorten the paper. In my opinion, it would be sufficient in this case to state and explain the expectations, and to provide no more than, say, 3-5 comparison figures and/or tables as proof. My feeling would be that it should be possible, without sacrificing scientific rigor, to condense the content for a target audience of aerosol modellers into a text that is at most half as long.

If the main goal is to describe an improvement of the model, I would still recommend some shortening (maybe even the same), but I would also request comparisons with observations, e.g., in analogy to the predecessor publication by Feng et al. (10.1029/2021MS002909). After all, an improvement in our simulation abilities can only be proven by better agreement with observations.

We thank the referee for their appreciation of our work and for the suggestion on being clear in scope and being concise in writing. We prefer to limit the scope of this paper to a discussion on numerical error without including comparisons with observations, because discrepancies between simulations and observations can result from multiple error sources including not only the numerical algorithms but also the formulation of the model equations, the values of the uncertain parameters, etc. The different types of errors might compensate each other, hence reducing one type of error might temporarily degrade the agreement with observations. We do note that the revised coupling scheme has been included in the candidate configuration of EAMv3; some of the colleagues on our author list will present comparisons with observations in a separate paper in the context of evaluating EAMv3. Therefore, in the revised version of this paper, we have chosen the first option suggested by the referee, i.e., to shorten the paper.

We have reduced the number of tables from 8 to 1 in the main body plus 1 in the supplementary material. We also reduced the number of figures from 12 in the main body plus 3 in the appendices to 9 in the main body and 3 in the supplementary material. When evaluating the impacts of the revised coupling, we now focus only on dust in the main body and briefly discuss the impacts on other species in the supplementary material. The paper is indeed more focused after these changes. Thanks again for the suggestion.

**Specific comments**

There are two questions that I would like to see addressed in addition to the current discussion (no more than a sentence or two necessary):

1. Could the problem also be tackled with a reduction of the model time step, or sub-time steps for the aerosol processes?

Yes. We added the following comment at the beginning of Sect. 3.3 before introducing the revised coupling scheme: The weaknesses of the original coupling scheme discussed in Sect. 3.2 are expected to have less impact on the model results if the overall model timestep is reduced or if the related aerosol processes are sub-cycled together so that the processes exchange information at shorter time intervals. However, both approaches would lead to significant increases in the computational cost of the model.

2. How do the differences between the simulations with the same vertical resolutions compare with interannual variability?

In terms of regional and annual averages, the differences in dust life cycle caused by the revision of coupling scheme in L72 simulations are substantially larger than the interannual variability. At specific locations, especially away from dust sources and where the wind variability is strong, the signal-to-noise ratio can be small. To demonstrate this, and also taking into account the referee's additional comments below on the nudging strategy and on decreases in the zonal mean POA mixing ratio in the original Fig. 12, we have decided to replace the 1-year nudged simulations presented in the original manuscript with 10-year free-running simulations of the years 2000 to 2009. In figures and tables in the revised manuscript where annual mean results are presented, we either show the interannual variability alongside the 10-year mean or mask out results where the differences between simulations are statistically insignificant according to the Kolmogorov-Smirnov two-sample test with a significance level of 0.05. All figures and tables in the revised manuscript have been updated.

In some places, I found the distinction between "source regions" and "non-source regions" a bit difficult, as I was unsure if or when the time dimension was included in this distinction, i.e., does a grid cell always belong to a source region, or only during time steps in which emissions actually occur?

In the revised manuscript, we have clarified that for the annual mean results, the identification of source and non-source regions was based on the annual mean emission fluxes of each year and each simulation (see Sect. 3.1, paragraph 2, and Sect. 4, end of paragraph 1). In the analyses of instantaneous model output, the identification of source and non-source grid columns was based on the instantaneous emission flux (Sect. 3.1, paragraph 4).

Some even more specific comments as a list:

- Abstract: "The revised scheme [...] better resembles the dust life cycle in the real world." -> This confused me in the beginning, as to me the "natural" sequence would seem to be first emissions, then mixing, then removal. Only much later I guessed from Fig. 1 that moving

mixing between emissions and removal would probably not be a simple task for coding reasons. This should be clarified here.

In response to this comment and a suggestion from referee 1, we have clarified in the revised abstract that the proposed revision is the one that requires the least amount of code changes.

- Introduction: The authors cite very few "state-of-the-art" aerosol-climate models in a few places. I suggest to either extend the reference lists in these places or cite a more general publication, like a book, a review or a model intercomparison paper.

Thanks for the suggestion. The revised manuscript cites the model intercomparison papers by Gliss et al. (2021, AeroCom III) and Myhre et al. (2013, AeroCom II).

- p. 3: "the revised coupling provides better results" -> I suggest to be more specific about the "better" here. It may refer to better in a numerical sense, which the companion paper shall demonstrate. If it is intended to refer to agreement with observations, I would request some evidence for this assertion.

We have changed the wording to "the revised coupling is more accurate in a numerical sense".

- p. 10: If nudging leads to a different surface wind climatology, is that not an argument _for_ the nudging, rather than against it? Furthermore, should the emissions not be driven by the "same" winds for this experiment?

Our understanding is that the statement "nudging leads to a different surface wind climatology" can be an argument against nudging winds *near the surface*, which is the reason why we applied nudging only above 850hPa in the original manuscript. Nudging in the upper layers can effectively constrain large-scale motions in the free troposphere and hence substantially suppress "noise" caused by synoptic variability.

That said, we agree with the referee's second point that since dust emissions are strongly sensitive to near-surface winds, the configuration presented in the original manuscript (which was inherited from some earlier studies focusing on ERFaer) was not the optimal one. Considering also the referee's question earlier about interannual variability and the comment below about the reductions in mixing ratio seen in parts of the zonal mean POA distribution, we decided to replace the 1-year wind-nudged simulations in the original manuscript with 10-year free-running simulations of the years 2000- 2009. All figures, tables, and discussions in the manuscript have been updated. In future studies aiming at further improving the numerical process coupling, we will probably conduct simulations that are nudged to the model's own meteorology in all vertical layers.

- Fig. 3: A comment on the discontinuities in the lowest 2 - 3 layers should be added.

These discontinuities are possibly non-local effects of the very strong discontinuities between the second lowest and lowest model layers shown in that same figure and in the figures of statistical distribution of instantaneous output. The comparison between results obtained with the original and revised schemes (Fig. 10 in the original manuscript and Fig. 8 in the revised version) shows that when the revised scheme is used, the process rate profiles are much smoother in the lowest few layers. We have added this comment to Sect. 4.1 in the revised manuscript.

- Sect. 3.1:

  o     What is the motivation for the selection of the time range for the histogram? Is this representative of the whole year?

  The particular choice of 90-day period was based on technical convenience: the instantaneous output was written to files that each contained 30 days, and we simply took the first 3 files of the year 2000. We have also analyzed other 90-day periods of the year and shorter time periods like 1-month, 1-day, or 1 timestep and plotted the corresponding emission histograms as well as box plots showing statistical distributions of instantaneous mixing ratios and tendencies. Although the instantaneous emission fluxes changed for these different selections of time range, large contrasts were robustly seen in the magnitudes of process rates between the source, vicinity, and remote regions and in different altitude ranges. We have added this discussion to the revised manuscript at the end of Sect. 3.1

  o     What is the motivation for the distinction between the three "portions" of the histogram, and for the values of their borders?

  The following explanation has been added to the revised manuscript: Given the wide range of emission fluxes seen in the histogram, we speculated and then confirmed that the column samples with very week emissions (i.e., near the left end of the histogram) exhibited properties in the simulated dust mixing ratios and process rates that were similar to the samples in the source-vicinity regions, and those properties changed gradually as the collection of column samples was shifted from the left to the right of the histogram. In order to show results from the source regions that are both representative and impactful in terms of their contribution to the total emission, the middle portion of the histogram was selected for the analysis shown in the next figure (which contains box plots showing statistical distributions of instantaneous values).

- Fig. 12: The reasons for the reductions seen in parts of the zonal mean POA distribution should be explained.

Our understanding is that those apparent reductions were noise caused by natural variability, as our 1-year nudged simulation did not constrain winds below 850 hPa. Figure S3 in the supplementary material in the revised manuscript is a figure similar to the old Fig. 12 but shows 10-year results from free-running simulations of 2000 to 2009. The new Fig. S3 does not indicate reductions of POA in terms of 10-year averages. The relative differences (in 10-year

averages) are small (less than 30%), similar to the 1-year results in the old Fig. 12. We applied the Kolmogorov-Smirnov two-sample test with a significance level of 0.05 to the free-running simulations and found the differences in zonal mean POA mixing ratios to be insignificant at most latitudes and altitudes.

In the interest of reproducibility, it might be worthwhile to publish the scripts to create the figures and table data along with the model output.

Following this suggestion, all simulations scripts and analysis scripts used for the revised manuscript have been published on Zenodo at https://doi.org/10.5281/zenodo.10371316

Technical corrections

- p. 1: "tuning parameters" -> I assume this refers to parameters in the emissions computations. This should be stated explicitly, as "tuning" may otherwise be understood as tuning the radiation balance of the model.

Thanks for pointing out the potential ambiguity. In the revised manuscript, we clarify that we mean "tuning parameters of emission strength".

- p. 7: The title of Sect. 2.2.3 could include "activation/resuspension".

Revised as suggested.

- p. 9: "tracers (not including water vapor) stored in the Fortran variable cam_in%cflx" -> These should be listed or described, see also comments by the first referee.

We have removed the mentioning of the Fortran variable cam_in%cflx following the first referee's suggestion. We also reorganized this paragraph. The revised version reads "The surface emission fluxes of aerosols and the net fluxes (emission minus turbulent dry deposition) of precursor gases are converted to mixing ratio tendencies in the lowest model layer. These tendencies are applied over a 30-minute timestep to update the corresponding mixing ratios."

- Table 3: I suggest to remove this table. If not, it might be useful to give the sequences of process names/abbreviations instead of, or in addition to, "Original" and "Revised".

The table has been removed as suggested.

- p. 10: "the Earth surface" -> "the Earth's surface" (again on p. 13)

Corrected.

- Fig. 3: As there is only one point with non-zero emissions, I suggest to remove the line "Emissions" from the plots in panels (c) - (h).

Done.

- p. 13: "The upper row is the results" -> "The upper row shows the results"

We have revised the panel layout in this figure to make the plots larger so that it is a bit easier to see the markers as well as the newly added lines showing interannual variability. The referee's suggestion has been addressed during the revision of the caption.

- p. 13: "orders stronger" -> "orders of magnitude stronger"

Corrected.

- p. 15: "turbulence mixing" -> "turbulent mixing"

Corrected.

- Fig. 7, in my opinion, is redundant after Fig. 2.

We agree the message delivered by Fig. 7 can be inferred from Fig. 2. Considering that the main purpose of our manuscript is to remind the readers of an overlooked topic and to explain the technical issue to colleagues who often focus primarily on the model physics instead of numerics, we believe Fig. 7 in the original paper provides a more direct depiction of the numerical consequence of the sequential splitting and thin model layer. Hence, we would like to keep the old Fig. 7, although we have merged it into Fig. 2 as a subfigure in the revised manuscript.

- p. 18: "motivates" -> "motivate"

The sentence has been revised and the grammar mistake was eliminated.

- Table 4:

  o There may be an error in the units specification for the first four data rows. Probably, the given numbers refer to a flux per unit area? If so, I suggest to integrate them over the respective areas for a more intuitive presentation (and to adapt the caption and Table 5 accordingly).

    Thanks for the thoughtful comment. The numbers originally shown were fluxes per unit area multiplied by the surface area of the entire globe. For the source regions and non-source regions, we should have used the respective areas instead. This has been

corrected in the revised manuscript, so we are now showing the integrals suggested by the referee.

o   Suggestion: the emissions could also be included here for completeness, and for direct comparison.

Emissions have been added as the first row of the updated table.

o   I stumbled upon "vertically integrated dry and wet removal". Is this the (net) removal from the lowermost layer, i.e., what actually leaves the model domain

In the revised table caption, we have further emphasized that the numbers are shown for the interstitial dust mass. For dry removal, a vertical integral over the entire model domain equals the net flux leaving the bottom of the atmosphere.  Regarding wet removal, the last paragraph in Sect. 4.1 explains that "For interstitial aerosols, the column-integrated wet removal rates indicate the net effect of below-cloud wet removal (which converts interstitial aerosols to precipitation-borne aerosols) and aerosol resuspension from evaporating precipitation (which converts precipitation-borne aerosols to interstitial aerosols)." So, a vertically integrated wet removal rate that is negative reflects a net conversion of interstitial dust mass to precipitation-borne dust mass.

- p. 23: "reference differences" -> "relative differences"

Corrected (and figured moved to supplementary material).

- Fig. A2: The portion of the histogram in Fig. 4 that corresponds to the data shown here should be specified.

The figure has been removed as we shortened the manuscript.

- The bibliography should be groomed. Some of the links include a double "https://doi.org/", for instance, and I also noticed a "n/a".

Thanks for the careful review. It seems the occurrences of double "https://doi.org/" were caused by many BibTex entries downloaded from journal websites containing "https://doi.org/" in the "doi" field. These have been corrected.

The reference with an "n/a" is no longer cited in the revised manuscript, as we now present 10-year free-running simulations instead of 1-year nudged runs. But we have updated our BibTex library for that entry. Thanks again for helping us realize these issues.

- For future publications, I suggest a replacement of rainbow colors by a color scale that can be interpreted more easily by people with color vision deficiency, e.g., "cividis"

(10.1371/journal.pone.0199239). (I noticed after writing this that Copernicus requested it already.)

Thanks for the suggestion. We have replaced the color maps in contour plots, used different marker styles in line plots, and removed explicit mentions of specific colors in the text.

---

## Author Response (AR2)

**Response to Editor's Request for Correction**

**Editor's note:**

Dear authors,

Thank you for the very thorough revisions of the manuscript in response to the reviewers' comments. This looks all good to me. As agreed by email, please replace the temporary link in the data availability section with the doi that you obtained.

Thank you,
Axel Lauer
(handling topical editor)

**Authors' response:**

Dear Editor,

We have replaced the temporary link by the Zenodo record number 10407375 (DOI: 10.5281/zenodo.10407375) and added the following reference:

Wan, H. and Zhang, K.: Compressed EAMv1 simulation output for evaluating two aerosol process coupling schemes, https://doi.org/10.5281/zenodo.10407375, 2023b.

Sincerely,

Hui Wan